# Teaching an Active Learner with Contrastive Examples

**Chaoqi Wang**
University of Chicago
chaoqi@uchicago.edu

**Adish Singla**
MPI-SWS
adishs@mpi-sws.org

**Yuxin Chen**
University of Chicago
chenyuxin@uchicago.edu

## Abstract

We study the problem of active learning with the added twist that the learner is assisted by a helpful teacher. We consider the following natural interaction protocol: At each round, the learner proposes a query asking for the label of an instance $x^q$, the teacher provides the requested label $\{x^q, y^q\}$ along with explanatory information to guide the learning process. In this paper, we view this information in the form of an additional *contrastive example* ($\{x^c, y^c\}$) where $x^c$ is picked from a set constrained by $x^q$ (e.g., dissimilar instances with the same label). Our focus is to design a teaching algorithm that can provide an informative sequence of contrastive examples to the learner to speed up the learning process. We show that this leads to a challenging sequence optimization problem where the algorithm's choices at a given round depend on the history of interactions. We investigate an efficient teaching algorithm that adaptively picks these contrastive examples. We derive strong performance guarantees for our algorithm based on two problem-dependent parameters and further show that for specific types of active learners (e.g., a generalized binary search learner), the proposed teaching algorithm exhibits strong approximation guarantees. Finally, we illustrate our bounds and demonstrate the effectiveness of our teaching framework via two numerical case studies.

## 1 Introduction

Active learning characterizes a natural learning paradigm where a learner *actively engages* in the learning process, and has demonstrated great success in both the educational domain [Prince, 2004] and the machine learning literature [Settles, 2012]. In the context of machine learning, active learning has traditionally been studied as the *active instance labeling* problem, where the goal of an active learner is to learn a concept by selectively querying the labels of a sequence of data points [Kosaraju et al., 1999; Dasgupta, 2004; Nowak, 2008; Balcan et al., 2006; Gonen et al., 2013; Hanneke and Yang, 2014; Yan and Zhang, 2017]. Recently, active learning has been investigated under richer interaction protocols, including learning from feature feedback [Poulis and Dasgupta, 2017; Dasgupta and Sabato, 2020] or from demonstrations [Chernova and Thomaz, 2014; Silver et al., 2012]. Typically, these works assume that the rich feedback obtained from the teacher is adversarial [Dasgupta and Sabato, 2020] or stochastic [Poulis and Dasgupta, 2017]. While these results have shown promising improvements in sample complexity by leveraging more expressive feedback, the assumptions of an adversarial or stochastic teacher could fall short in characterizing many real-world interactive learning systems (e.g. automated tutoring or multi-agent learning systems) where the teacher cooperates with the learner to achieve the learning objective [Zilles et al., 2011; Zhu et al., 2018].

Motivated by real-world cooperative learning and teaching scenarios [Chen et al., 2018a; Teso and Kersting, 2019], we consider an optimistic yet natural interaction protocol between an active learner and a *helpful* teacher (see Figure 1 (c)): At each round, the learner proposes a query asking for the label of an instance $x^q$; the teacher then provides the requested label $\{x^q, y^q\}$ along with explanatory

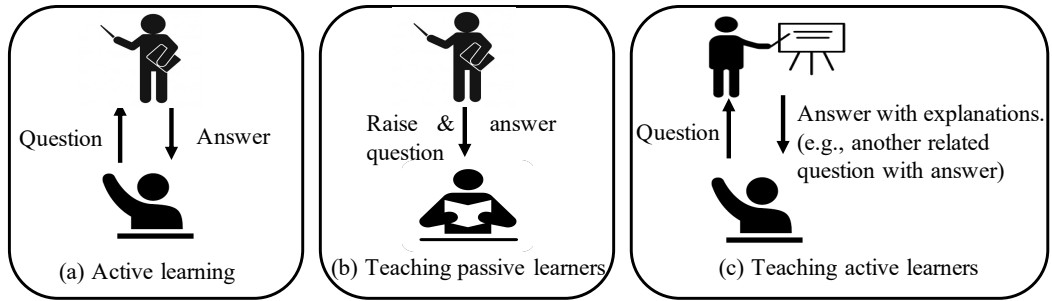

**Figure 1:** The difference between (a) active learning, (b) classic machine teaching and (c) our problem.

information to guide the learning process. In this paper, we consider this information in the form of an additional *contrastive example* ($\{x^c, y^c\}$) where $x^c$ is picked from a set constrained by $x^q$ (e.g., dissimilar instances with the same label, or similar instances with opposite labels). In contrast to existing work in active learning with contrastive examples [Abe et al., 2006; Liang et al., 2020; Ma et al., 2021] or with comparison queries [Jamieson and Nowak, 2011; Kane et al., 2017] which focus on developing *active learning algorithms*, we assume that the learner's querying and learning strategy is *fixed* and *known* to the teacher, and focus on designing a *teaching algorithm* that can provide an informative sequence of contrastive examples to the learner to speed up the learning process.

Note that in the absence of the learner's query, this problem reduces to the classical *machine teaching* problem [Goldman and Kearns, 1995]. When teaching a (passive) version space learner [Goldman and Kearns, 1995], classical machine teaching can be viewed as a set cover problem, and the greedy algorithm is known to compute a cover that is within a logarithmic factor of the minimum cover [Chvatal, 1979]. However, as we show in section 3.3, an active learner actively trying to reduce the label cost could actually *hurt* the performance of a greedy teacher.

Given an active version space learner specified by an acquisition function, we show that the design of an optimal teaching algorithm induces a challenging sequence optimization problem, where the teacher's available choices of contrastive examples at a given round depend on the history of interactions. We study the greedy teacher that adaptively picks contrastive examples in response to the learner's queries. Specifically, we introduce several natural problem-dependent parameters to characterize (1) the constraints imposed by the active learner and (2) the structure of the coverage function resulting from the hypothesis space. Based on these characterizations, we explore the theoretical properties for the greedy teacher, and demonstrate the power and limitations of the interactive teaching framework. Our main technical contributions are summarized are:

I. We consider a novel interactive protocol for teaching an active learner, and derive a general performance guarantee for the greedy teacher, regardless of the choice of acquisition function by the active learner. Our result generalizes existing near-optimality guarantees for the greedy teacher under the classical teaching [Goldman and Kearns, 1995] to the more challenging sequence optimization problem (section 3).

II. We then show that, perhaps surprisingly, a greedy teacher may fail badly when interacting with an active learner compared to the optimal teacher, due to the constrained choices of contrastive examples imposed by the learner's query (section 3). In fact, there exist pessimistic scenarios where an active learner alone is better off without the help of a greedy teacher (section 4).

III. When interacting with a generalized binary search (GBS) learner [Dasgupta, 2004; Nowak, 2008], we provide a strong performance guarantee for the greedy teacher, and show that it cannot perform arbitrarily badly (section 4). Our bound establishes a rigorous connection between the problem-dependent parameters introduced in section 3 to the classical *k-neighborly condition* and *coherence parameter* which are widely used in the active learning literature [Nowak, 2008, 2011; Mussmann and Liang, 2018].

IV. We empirically illustrate our bounds via two experiments. These examples illustrate the intuitive sequential teaching process and demonstrate the effectiveness of our teaching framework.

## 1.1 Related Work

**Teaching a sequential learner** Machine teaching has shown significant development in the past two decades in both theoretical and practical domains. In a quest to achieve rich teacher-learner interactions, various different models for machine teaching have been proposed under the sequential teaching setting (e.g., local preference-based model for version space learners [Chen et al., 2018b], models for gradient learners [Liu et al., 2017, 2018; Kamalaruban et al., 2019], models inspired by control theory [Zhu, 2018; Lessard et al., 2019], models for sequential tasks [Cakmak and Lopes, 2012; Haug et al., 2018; Tschiatschek et al., 2019], and models for human-centered applications that require adaptivity [Singla et al., 2013; Hunziker et al., 2019]). Compared to our setting, in these works, the teacher is responsible for designing all of the learning data. Recently, Peltola et al. [2019] investigated the task of teaching sequential learners such as multi-armed bandits or active learners where the learners also pose query, but they formulate the problem as a variant of reward shaping, and allow the teacher to provide teaching examples that are inconsistent with the true data distribution. While our work falls into the category of teaching sequential learners, it could be viewed as a form of explanation-based machine teaching, and hence inherits different teaching constraints and objectives from the above settings.

**Explanation-based teaching and learning** Explanation-based teaching captures a rich class of teaching protocols where labels are coupled with additional information (such as highlighting regions or features on an image). Having an additional mode in the feedback allows the learner to perceive more intuitive explanations of the target hypothesis, and hence dramatically improves the learner's ability to learn a new concept. Explanations were shown to be effective in helping a human student to improve classification performance by steering the student's attention [Grant and Spivey, 2003; Roads et al., 2016; Chen et al., 2018a; Aodha et al., 2018]; likewise, they also play an important role in helping machine learners to efficiently learn some concept classes that would otherwise form intractable learning problems [Poulis and Dasgupta, 2017; Dasgupta and Sabato, 2020]. Our work is motivated by the success of explanation-based learning, and consider contrastive examples as a simple yet effective form of explanations. In comparison with existing work on explanation-based teaching, our work has an additional twist featured by the interaction with an active learner.

**Submodular set and sequence optimization** Our theoretical framework in section 3 is inspired by recent results on sequence submodular function maximization [Zhang et al., 2015; Tschiatschek et al., 2017; Hunziker et al., 2019]. In particular, Zhang et al. [2015] introduced the notion of string submodular functions, which, analogous to the classical notion of submodular set functions [Krause and Golovin, 2014], enjoy similar performance guarantees for maximization of deterministic sequence functions. Our setting has three key differences in that (1) we focus on a constrained planning problem where the set of available actions is dynamic (i.e. depending on the learner's query), (2) our objective function no longer exhibits diminishing returns properties and (3) we consider a min-cost coverage problem as opposed to the budgeted maximization problem.

## 2 Active Learning with a Teacher: Problem Formulation

In this section, we formally introduce the new problem of active learning with a helpful teacher along with the corresponding interaction protocol. Then, we formulate the problem of teaching an active learner as a constrained sequential optimization problem. Throughout the entire paper, we use $\mathcal{X}$ to denote the ground set of unlabeled instances. We consider a finite class of hypothesis $\mathcal{H}$, where each hypothesis $h \in \mathcal{H}$ is a binary classifier with its output being either $+1$ or $-1$ (i.e., $h(x) \in \{+1, -1\}$).

### 2.1 Learner-Teacher Interaction Protocol

We study the problem of teaching an active version space learner with a teacher. In contrast to the standard active learning protocol, our framework not only allows the learner to ask questions but also enables the teacher to provide additional explanatory information to guide the learning process. Specifically, we model the explanatory information using a contrastive example picked from a set constrained by the learner's query (e.g., dissimilar instances with the same label, or similar instances with opposite labels). The full interaction protocol is presented in Protocol 1. At each round $t$, the learner asks a query $x_t^{\mathrm{q}} = q(x_{1:t-1}^{\mathrm{c}}, x_{1:t-1}^{\mathrm{q}}, H_{t-1}^{q})$ with query function $q$ conditioning on the received

---

**Protocol 1** An interaction protocol between the teacher and the active learner

---

1: An active learner with a query function $q$; Initial version space $H_0^q = \mathcal{H}$; Constraint function $\xi$ for the contrastive example; Target hypothesis $h^\star$.
2: **for** $t = 1, 2, 3, \ldots$ **do**
3:   learner asks a query $x_t^{\mathrm{q}} = q(x_{1:t-1}^{\mathrm{c}}, x_{1:t-1}^{\mathrm{q}}, H_{t-1}^q)$
4:   teacher provides the label $y_t^q$ for the query $x_t^{\mathrm{q}}$
5:   teacher provides a contrastive example $(x_t^{\mathrm{c}}, y_t^c)$ from a constrained set $\xi(x_t^{\mathrm{q}})$.
6:   learner computes the updated version space $H_t^q$.
7:   **if** $H_t^q = \{h^\star\}$ **then**  teaching process terminates.

---

contrastive examples $(x_{1:t-1}^{\mathrm{c}})$ and version space $H_{t-1}^q$. Then the teacher returns the ground-truth label $y_t^{\mathrm{q}}$ for $x_t^{\mathrm{q}}$ and also provides a contrastive example $(x_t^{\mathrm{c}}, y_t^c)$. Under this setting, the version space at iteration $t$ with teaching sequence $x_{1:t}^{\mathrm{c}} = \{x_i^{\mathrm{c}}\}_{i=1}^t$, can be computed by (we omitted $y$ for clarity)

$$H_t^q = H^q(x_{1:t}^{\mathrm{c}}) = \mathcal{H} \setminus \left( \bigcup_{i=1}^t S(x_i^{\mathrm{q}}) \cup S(x_i^{\mathrm{c}}) \right), \tag{1}$$

where $\mathcal{H}$ is the initial version space of the active learner and $S(x) = \{h \in \mathcal{H} | h(x) \neq h^\star(x)\}$ denotes the hypotheses covered by $x$ (i.e., the hypotheses that predict different labels on $x$ than $h^\star$).

## 2.2 Optimization Objective

Our goal is to reduce the learner's version space such that only $h^\star$ (the target hypothesis) is left with minimal cost (e.g., # of interaction rounds or # of samples). The definition of cost can be problem-dependent. For the general case, we consider the following form of the cost function

$$J^q(x_{1:t}^{\mathrm{c}}) = \sum_{i=1}^t c(x_i^{\mathrm{c}}) + c(x_i^{\mathrm{q}}), \tag{2}$$

where $x_{1:t}^{\mathrm{c}}$ is the teaching sequence given by the teacher, and $c : \mathcal{X} \to \mathbb{R}_+$ computes the cost of the example. We don't write the dependencies on the learner's query sequence explicitly, as the queries are induced by the teaching sequence and the query function $q$ accordingly.

Given the active learner, we seek the optimal teaching sequence that achieves minimal cost by solving the following optimization problem.

$$\min_{X \in \mathcal{X}^\star} J^q(X) \qquad s.t. \qquad H^q(X) = \{h^\star\} \quad \text{and} \quad x_t \in \xi(x_t^{\mathrm{q}}) \ \forall t \in [|X|], \tag{3}$$

where we use $\mathcal{X}^\star$ to denote the set of all possible sequences of teaching examples, and $[n]$ denotes all the integers in $[1, n]$. At every iteration, we can compute the constraint set by the function $\xi : \mathcal{X} \to 2^{\mathcal{X}}$ given the learner's query. Then, the teacher must select the contrastive example from the constrained set. This is a natural consideration, since without the constraint, the teacher might have selected some contrastive examples that are not interpretable to the learner. In practice, the constrained set could be the instances that are *dissimilar* to the learner's query but with the *same* label or the instances that are *similar* to the learner's query but with *different* label.

To be noted, since the order of the teaching sequence matters in our problem, it expands the search space exponentially. This makes solving the above sequence optimization problem far more difficult than solving the corresponding set optimization problem (e.g., classical machine teaching problem), which is NP-hard. Hence, we turn to approximation algorithms with polynomial time complexity.

## 3 Greedy Teaching Algorithm and Analysis: A General Complexity Bound

In this section, we propose a simple yet effective greedy algorithm for solving the sequential optimization problem defined in the last section. Then, we further provide theoretical results to upper and lower bound the number of rounds of the interaction between the learner and the teacher or the sample complexity, for the general cases (i.e., we don't have any assumptions on the active learner).[1]

---

[1] All the proofs are deferred to the Appendix due to page limit.

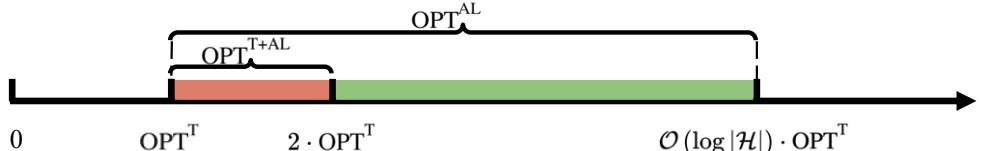

**Figure 2:** Relationship between the teaching dimension ($\text{OPT}^{\text{T}}$), the optimal sample complexity for active learning ($\text{OPT}^{\text{AL}}$), and the sample complexity for active learning with an optimal teacher ($\text{OPT}^{\text{T+AL}}$).

---

**Algorithm 2** Greedy teaching algorithm for active leaner

---

1: An active learner with a query function $q$; Initial version space $H_0^q = \mathcal{H}$; Constraint function $\xi$ for the contrastive example; Target hypothesis $h^\star$.
2: **for** $t = 1, 2, 3, \ldots$ **do**
3:     learner asks a query $x_t^q = q(x_{1:t-1}^c, x_{1:t-1}^q, H_{t-1}^q)$.
4:     teacher provides the label $y_t^q = h^\star(x_t^q)$.
5:     teacher provides a contrastive example $x_t^c = \arg\max_{x \in \xi(x_t^q)} \Delta(x|x_{1:t-1}^c, q, \mathcal{H})$ (equation 5).
6:     Update learner's version space by equation 1.
7:     **if** $H_t^q = \{h^\star\}$ **then** teaching process terminates.

---

### 3.1 Analysis on the Optimal Teacher

To demonstrate the effectiveness of our framework, we first conduct an analysis on the relationship between the teaching dimension ($\text{OPT}^{\text{T}}$), the optimal sample complexity for active learning ($\text{OPT}^{\text{AL}}$), and the sample complexity for active learning with an optimal and *unconstrained* teacher ($\text{OPT}^{\text{T+AL}}$).

First of all, it is not too hard to show that $\text{OPT}^{\text{T+AL}} \leq \text{OPT}^{\text{AL}}$. This result can be proved by the fact that the optimal teacher cannot perform worse than such a teacher, which mimics the query selection strategy of the active learner. More importantly, this result also implies that the optimal teacher is always helpful for the active learner. In addition, by the following Lemma 1, we have that $\text{OPT}^{\text{T}} \leq \text{OPT}^{\text{T+AL}} \leq 2 \cdot \text{OPT}^{\text{T}}$, which lower and upper bounds $\text{OPT}^{\text{T+AL}}$.

**Lemma 1.** *For any $(\mathcal{X}, \mathcal{H})$ and active learner with query function $q$, we have $OPT^T \leq OPT^{T+AL} \leq 2 \cdot OPT^T$, where $OPT^T$ is the classic teaching complexity.*

Lastly, to further understand the relationship between $\text{OPT}^{\text{T+AL}}$ and $\text{OPT}^{\text{AL}}$, we borrow the results from Hanneke [2007]. As proved in Hanneke [2007], we have $\text{OPT}^{\text{T}} \leq \text{OPT}^{\text{AL}} \leq \log |\mathcal{H}| \cdot \text{OPT}^{\text{T}}$. Therefore, $\text{OPT}^{\text{AL}}$ can potentially be as large as $\log |\mathcal{H}| \cdot \text{OPT}^{\text{T}}$, while $\text{OPT}^{\text{T+AL}}$ is upper bounded by $2 \cdot \text{OPT}^{\text{T}}$. This distinction shows that the gap between $\text{OPT}^{\text{T+AL}}$ and $\text{OPT}^{\text{AL}}$ can be arbitrarily large, hence demonstrates the advantage of incorporating the teacher in active learning. A visual illustration is provided in Figure 2. Throughout the entire paper, we use $\text{OPT}^{\text{T+AL}}$ to denote the sample complexity of active learning with an initial hypothesis class $\mathcal{H}$, and an optimal and unconstrained teacher.

### 3.2 Greedy Teaching Policy

We now present a simple yet effective greedy teaching algorithm for teaching an active learner (see Algorithm 2). For clarity of presentation, we focus on unit cost case, i.e., $c(x) = 1$ for all $x$, which is equivalent to minimizing the number of rounds of interactions between the teacher and active learner. The extension to a more general cost will be discussed in the Appendix.

Therefore, to obtain the teaching sequence, we will solve the following minimum-cost cover problem

$$\min_{X \in \mathcal{X}^\star} |X| \quad s.t. \quad H^q(X) = \{h^\star\} \quad \text{and} \quad x_t \in \xi(x_t^q) \;\; \forall t \in [|X|]. \tag{4}$$

For iteration $t + 1$ with teaching history $x_{1:t}^c$, we define the marginal gain as the following

$$\Delta(x^c|x_{1:t}^c, q, \mathcal{H}) = f(x_{1:t}^c \oplus x^c|q, \mathcal{H}) - f(x_{1:t}^c|q, \mathcal{H}) = |H^q(x_{1:t}^c) \setminus H^q(x_{1:t}^c \oplus x^c)|, \tag{5}$$

where $f(x_{1:t}|q, \mathcal{H}) = |\mathcal{H} \setminus H^q(x_{1:t})|$ is the number of hypotheses removed from the version by the teaching examples $x_{1:t}$ together with the corresponding learner's queries (defined by equation 1), and

$\oplus$ denotes the concatenation between two sequences. To be noted, when the function $f$ is maximized, it is equivalent to that only the target hypothesis $h^\star$ is left in the version space.

For the greedy teaching algorithm, as described in Algorithm 2, at each round, it selects the contrastive example from the constrained set that maximizes the marginal gain defined in equation 5. Such process will be repeated until the function $f$ is maximized or the budget is used up.

## 3.3 Theoretical Guarantees for Greedy Teaching Policy

We now turn to analyzing the theoretical performance of the greedy algorithm. In general, for active learning, there is no guarantee for the sample complexity of arbitrary learners, and sometimes, the worst case complexity can be up to $\mathcal{O}(|\mathcal{H}|)$ [Dasgupta, 2004], which is uninformative.

To understand the role of the greedy teacher in active learning, we rely on the following two properties of the objective function $f$. These two properties are usually used in the characterizing the submodularity or string submodularity of a function [Zhang et al., 2015; Hunziker et al., 2019].

**Definition 1** (**Pointwise submodularity ratio**). *For any sequence function $f$, the pointwise submodularity ratio with respect to any sequences $\sigma$, $\sigma'$ and query function $q$ and hypothesis class $H$ is defined as*

$$\rho_H(\sigma, \sigma', q) = \min_{x \in \mathcal{X}} \frac{\Delta(x|\sigma, q, H)}{\Delta(x|\sigma \oplus \sigma', q, H)}. \tag{6}$$

**Definition 2** (**Pointwise backward curvature**). *For any sequence function $f$, the pointwise backward curvature with respect to any sequences $\sigma$, $\sigma'$ and query function $q$ and hypothesis class $H$ is defined as*

$$\gamma_H(\sigma, \sigma', q) = 1 - \frac{f(\sigma' \oplus \sigma|q, H) - f(\sigma|q, H)}{f(\sigma'|q, H) - f(\emptyset|q, H)}. \tag{7}$$

The submodularity ratio measures the degree of diminishing returns of the underlying sequence function by computing the ratio between the marginal gains. When the sequence function $f$ is string/sequence submodular [Zhang et al., 2015; Alaei et al., 2010], the minimum pointwise submodularity ratio over all possible sequences is 1. By contrast, the pointwise backward curvature measures the degree of diminishing returns of the difference between marginal gains. For postfix monotone sequence functions, the pointwise backward curvature is no greater than 1 [Zhang et al., 2015].

In addition, at each round, the teacher can only select examples from a constrained set, which limits its power. To accommodate such limitations, we can view the teacher as an $\alpha$-approximate greedy teacher in the analysis. The parameter $\alpha$ characterizes the ratio between the marginal gain of an unconstrained greedy teacher and that of the constrained greedy teacher. For an active learner with query function $q$, constraint $\xi$, and version space $H_0^q = \mathcal{H}$, we can compute the corresponding $\alpha$ by

$$\alpha = \max_t \frac{\max_{x \in \mathcal{X}} \Delta(x|x_{1:t-1}^c, q, \mathcal{H})}{\max_{x \in \xi(x_t^q)} \Delta(x|x_{1:t-1}^c, q, \mathcal{H})}. \tag{8}$$

In general, the parameter $\alpha$ measures how strict the constraint is. When $\alpha$ becomes large, the effectiveness of the greedy teacher will correspondingly decrease.

The following theorem summarizes our first theoretical result, which provides a problem-dependent upper bound for the sample complexity of greedy teaching algorithms.

**Theorem 1.** *The sample complexity of the $\alpha$-approximate greedy algorithm for any active learner with a initial hypothesis class $\mathcal{H}$ is at most*

$$\left( \frac{\alpha \cdot \mathcal{O}\left(\log |\mathcal{H}| \cdot \log\left(|\mathcal{H}|/\gamma^g\right)\right)}{\rho^g \gamma^g \log(\gamma^g/(\gamma^g - 1))} + \frac{\alpha \cdot \mathcal{O}\left(\log\left(|\mathcal{H}|/\gamma^g\right)\right)}{\rho^g \gamma^g} \right) \cdot OPT^{T+AL}, \tag{9}$$

*where $\gamma^g = \max_{H \in \mathcal{H}'} \gamma_H^g$ and $\rho^g = \min_{H \in \mathcal{H}'} \rho_H^g$ with*

$$\gamma_H^g = \max_{i \geq 1} \gamma_H(\sigma^H, x_{1:i}^H, q), \quad \rho_H^g = \min_{i,j \geq 0} \rho_H(x_{1:i}^H, \sigma_{1:j}^H, q), \tag{10}$$

*which are computed with respect to the greedy teaching sequence $x^H$ and the optimal teaching sequence $\sigma^H$ for the corresponding active learner with a initial hypothesis class $H \in \mathcal{H}' = \{H^q(X)|X \in \mathcal{X}^\star \wedge |H^q(X)| \geq 2\}$ and query function $q$.*

**Remark 1.** *For any active learner, $1 \leq \gamma^g \leq |\mathcal{H}| - 1$ and $\rho^g \leq 1$. When $\gamma \to 1$, the l.h.s. term in the parentheses of equation 4 will vanish. When $\gamma^g \to |\mathcal{H}| - 1$, the r.h.s. term will vanish.*

Theorem 1 bounds the worst case sample complexity of greedy teaching algorithms. To be noted, both $\gamma^g$ and $\rho^g$ are computed with respect to the greedy teaching sequence and the optimal teaching sequence for the active learner with some initial hypothesis class $H \subseteq \mathcal{H}$, rather than two arbitrary sequences. Therefore, based on Remark 1 and Zhang et al. [2015], if the objective function is postfix monotone, then we will have $\gamma^g = 1$, otherwise, we can only conclude that $\gamma^g \geq 1$. In particular, when $\gamma^g \to 1$, the r.h.s. term in equation 4 will dominate, whereas when $\gamma^g \to |\mathcal{H}| - 1$, the l.h.s. term will become dominating. Informally, $\gamma^g$ can be a measure of the difficulty of the underlying sequence optimization problem, and a larger $\gamma^g$ will be likely to imply that the problem is more difficult. In what follows, we further argue that the linear dependency on $\alpha$ cannot be avoided, showing that Theorem 1 is non-vacuous.

**Theorem 2.** *There exists a constraint function $\xi$, version space $\mathcal{H}$ and active learner with query function $q$, such that the sample complexity of the greedy teacher is at least $\Omega(\sqrt{|\mathcal{H}|}) \cdot OPT^{T+AL}$.*

Theorem 2 shows that the linear dependency on $\alpha$ cannot be improved when without any assumptions on the constraint function $\xi$, hypotheses space and the active learner. To porve Theorem 2, we constructed a special case, detailed in the Appendix. In the example, the optimal teacher only uses a constant number of examples to teach the active learner, while the greedy teacher needs to provide all the examples, whose number is of the order $\Omega(\sqrt{|\mathcal{H}|})$.

## 4 Teaching Greedy Active Learners

In this section, we consider the problem of teaching a greedy active learner, concretely, the Generalized Binary Search (GBS) [Nowak, 2008], which selects the query in a greedy manner (i.e., the one that maximizes the utility). At each iteration $t + 1$ with the version space $H_t$, the GBS learner selects the query according to the expected number of hypotheses that can be covered by the query (i.e., the utility of the query). Specifically, for binary classification problem, for each query $x$, the utility can be computed as

$$u(x|H_t) = \frac{2}{|H_t|} \cdot \left( \sum_{h \in H_t} \mathbb{1}[h(x) = -1] \right) \cdot \left( \sum_{h \in H_t} \mathbb{1}[h(x) = 1] \right). \tag{11}$$

The above utility function measures how discriminative the selected query is. GBS selects the query that maximizes the utility, and hence it lower bounds the minimal number of hypotheses to be removed after the query. As pointed out in Dasgupta [2004], the worst case sample complexity for the GBS learner is $\mathcal{O}(|\mathcal{H}|)$. However, for practical problems, there are usually some structures in the hypotheses space and the sample space we can exploit to get a tighter bound. Specifically, we consider problems with the $k$-neighborly structure [Nowak, 2008; Mussmann and Liang, 2018].

**Definition 3** ($k$-**neighborly**). *Consider the graph $(V, E)$ with vertex set $V = \mathcal{X}$, and edge set $E = \{(x, x')|d_{\mathcal{H}}(x, x') \leq k, \forall x, x' \in \mathcal{X}\}$, where $d_{\mathcal{H}}(x, x') = |\{h|h \in \mathcal{H} \text{ and } h(x) \neq h(x')\}|$. The query and hypotheses space $(\mathcal{X}, \mathcal{H})$ is $k$-neighborly if the induced graph is connected.*

Intuitively, the $k$-neighborly structure ensures that for any two points $x, x' \in \mathcal{X}$, there is a path formed by similar pairs connecting them. With such structures, Nowak [2011] further introduces the *coherence parameter*, which plays an important role in bounding the worst case complexity of GBS.

**Definition 4** (**Coherence parameter**). *The coherence parameter for $(\mathcal{X}, \mathcal{H})$ is defined as*

$$c^\star(\mathcal{X}, \mathcal{H}) := \min_P \max_{h \in \mathcal{H}} \left| \sum_{x \in \mathcal{X}} h(x) dP(x) \right|, \tag{12}$$

*where we minimize over all possible distributions on $\mathcal{X}$.*

The coherence parameter $c^\star$ measures how evenly the query selected by GBS bisects the hypotheses. With the above assumptions, Nowak [2008] shows that the sample complexity of GBS can be upper bounded by $\mathcal{O}(\log(|\mathcal{H}|)/\log(1/\eta))$, where $\eta = \max\{(1 + c^\star)/2, (k + 1)/(k + 2)\}$. This implies that, the smaller $c^\star$ and $k$, the more efficient GBS is.

Next, we seek to understand the sample complexity when pairing GBS with a greedy teacher. Can we also upper bound its sample complexity, even though the interaction between the learner and the teacher introduces extra complexity? The result is summarized in the following theorem.

**Theorem 3.** *For the GBS learner with a initial hypothesis class $\mathcal{H}$ and ground set $\mathcal{X}$, if $(\mathcal{X}, \mathcal{H})$ is $k$-neighborly and with coherence parameter $c^\star$, then the sample complexity of the greedy teaching algorithm with any constraint function $\xi$ is at most*

$$\frac{\alpha}{\epsilon} \cdot \mathcal{O}\left(\log^2(|\mathcal{H}|)\right) \cdot OPT^{T+AL}, \tag{13}$$

*where $\epsilon = \min\left\{(1-c^\star)/(1+c^\star), c^\star/(k-c^\star)\right\}$, and $\alpha \leq \max\{k/c^\star, 2/(1-c^\star)\}$.*

Theorem 3 gives a near-optimal bound for the greedy teaching algorithm for GBS learners, which guarantees that its performance cannot be arbitrarily bad. However, the bound cannot be directly compared to that of GBS alone, as we don't know $OPT^{T+AL}$. Unfortunately, our following remark shows that the sample complexity of GBS with a greedy teacher is not guaranteed to be smaller compared to GBS alone, and for some extreme cases, it can be even larger.

**Remark 2.** *The sample complexity of GBS with greedy teacher (even without the constraints) is not guaranteed to be smaller than that of GBS alone.*

The bad news from Remark 2 shows that the greedy teacher may not be satisfactory for teaching a GBS learner under some circumstances, though the greedy teaching algorithm performs very well in classic machine teaching problems. For a better understanding of the failure of greedy teachers, we provide a realizable example (see Appendix), where the GBS with a greedy teacher will requires more samples than that of GBS alone. We hope that this example can motivate improved teaching algorithms for active learners, beyond the greedy heuristics.

## 5  Numerical Experiments

In this section, we empirically evaluate the greedy teaching algorithm with $\beta$-greedy active learners (a rich class of active learners including GBS) in a simulated environment. We seek to understand how the type of active learner (i.e., varying $\beta$) and the constraint on contrastive examples affect the performance of the greedy teacher. To dig deeper, we further demonstrate how these two factors affect the value of $\alpha$ and $\rho^g$, which plays an important role in our theoretical results.

### 5.1  Experimental Setup

$\beta$**-greedy active learners:** In the experiments, we consider the class of $\beta$-greedy active learners, for which the selected query $x_{t+1}^q$ satisfies

$$u(x_{t+1}^q|H_t) \geq \frac{1}{\beta} \cdot \max_{x \in \mathcal{X}} u(x|H_t). \tag{14}$$

We can interpolate $\beta$ from 1 to $+\infty$ to cover a broad range of active learners with different properties. Specifically, when $\beta = 1$, we recover the GBS active learner, whereas, when $\beta = +\infty$, we recover the random active leaner. In general, as $\beta$ increases, the strategy becomes less greedy. For the $\beta$-greedy active leaner, the query function $q$ returns a query that satisfies inequality (14).

**Datasets:** We conducted experiments on two datasets: `Vespula-Weevil` and `Butterfly-Moth`, which were adopted by Singla et al. [2013, 2014] for verifying the algorithm for teaching the crowd to classify images. Specifically, `Vespula-Weevil` is a synthesized dataset with 200 images, which are generated by sampling 2-$d$ features from two bi-variate Gaussian distributions. The dataset `Butterfly-Moth` consists of 160 real images with the two categories Butterfly and Moth. The 2-d embedding of each image is inferred from the annotation process using a generative Bayesian model proposed in Welinder et al. [2010]. More details about the datasets can be found in the Appendix.

**Hypothesis space:** Following Singla et al. [2014], for the `Vespula-Weevil` dataset, we generate the hypotheses by randomly sampling from eight multivariate Gaussian distributions, with mean $\boldsymbol{\mu}_i = [\pi/4 \cdot i, 0]$, and covariance $\boldsymbol{\Sigma}_i = [2, 0; 0, 5e^{-3}]$, where $i$ is varying from 0 to 7. For the `Butterfly-Moth` dataset, the hypotheses are inferred from the annotation process with the same generative Bayesian model for inferring the 2-d embedding.

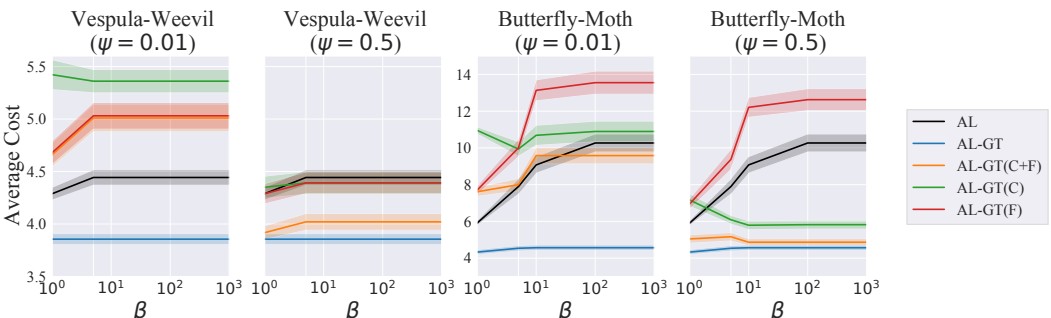

**Figure 3:** The average cost for different active learners and constraints. **AL** denotes the setting of active learner alone. **AL-GT** corresponds to active learner with an unconstrained greedy teacher, and **C, F, C+F** correspond to different constraints on the constrastive examples. $\psi \in [0, 1]$ is a parameter controlling the size of the search space, which only affects **AL-GT(C), AL-GT(F)**, and **AL-GT(C+F)**. The shaded area is the one-standard error.

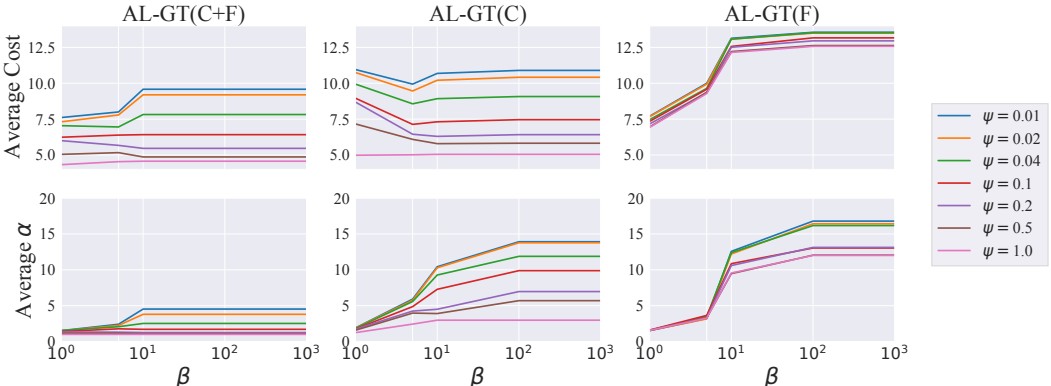

**Figure 4:** How do active learners ($\beta$) and constraints ($\psi$) affect the teaching cost (top row) and $\alpha$ (bottom row). The three columns are corresponding to the results of **AL-GT(C+F)**, **AL-GT(C)** and **AL-GT(F)** respectively. The results are obtained on the `Butterfly-Moth` dataset. We don't display the standard error for clarity.

## 5.2 Effects of Active Learners and Constraints

We first study how does the active learner (AL) affects the teaching performance under different constraints. Specifically, we vary $\beta$ in $[1, 5, 10, 100, 1,000]$ to model different learners. In terms of the constraint on the teacher's example, we consider the following three types of the constraints: 1) examples that are close to the learner's query but with different label (denoted by **C**); 2) examples that are far away from the learner's query but with the same label(denoted by **F**); 3) the union of 1)& 2) (denoted by **C+F**). For each constraint, we also adopt a parameter $\psi \in [0, 1]$ to control the size of the search space (i.e., the closest or furthest $\psi$ portion of the points with different or the same label as the query). We also include the unconstrained teacher, which can freely select examples.

To obtain more robust results, we iterate through all the hypotheses to serve as the target hypothesis once, and compute the averaged cost for identifying the target hypothesis, accompanied by the standard error. The results are presented in Figure 3. We observe that for both datasets, AL with greedy teacher (AL-GT, unconstrained) always achieves the best performance in terms the average cost. When $\beta$ becomes larger, the average cost of AL-GT increase very slightly. For the others, when $\psi$ is small, they usually perform worse than AL along. This is because when $\psi$ is small, the candidate points in the constrained set are too restrictive and they are usually redundant to the query, making the contrastive example uninformative. By contrast, when $\psi$ is large enough, AL-GT(C+F) can significantly outperform AL alone, showing that the teacher is helpful in this case. In summary, when $\beta$ becomes large and the constraint is not too strict, the teacher will be more helpful (i.e., the gap between AL and AL with teacher becomes larger).

We further study how the constraints and $\beta$ affect the parameters $\alpha$. The results are provided in Figure 4. In general, we can observe that as the constraints become stricter (i.e., $\psi$ becomes smaller),

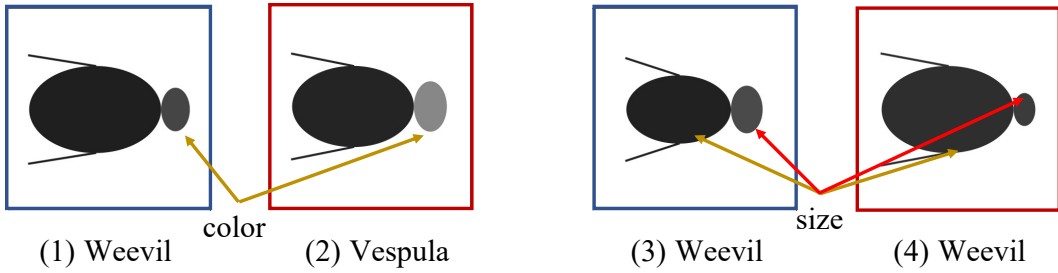

**Figure 5:** A visualization of a teaching sequence. (1)&(3) are learner's queries and (2)&(4) are the contrastive examples provided by the teacher (i.e., AL-GT(C+F)).

$\alpha$ will correspondingly increase as well as the cost for teaching the active learner. This is consistent with our Theorem 1 and Theorem 3 in Section 3 and Section 4. In addition, when $\beta$ becomes larger, $\alpha$ is also increasing, which is consistent with the trend in the cost plots (top row in Figure 4). Besides, we also compute the empirical values of $\rho^g$, finding that for different $\beta$ and constraints, $\rho^g$ stays very close to 1. According to our theoretical bounds, this may also explain why the trends in the cost plots are mostly consistent with those in the plots of $\alpha$ (bottom row in Figure 4).

### 5.3 Visualization of the Teaching Sequence

Lastly, we visualize a teaching sequence of AL-GT(C+F) in Figure 5. The results are obtained with $\beta = 1$ and $\psi = 0.5$ on the `Vespula-Weevil` dataset. The first and third examples are the learner's queries and the second and fourth ones are the teacher's provided contrastive examples. In general, weevils have relatively short heads with similar color to the body. In contrast, vespula have big and contrasting heads. From Figure 5, we can observe that at the first round, the teacher selects an image of vespula with similar body size to the learner's query, but with a head that is in relatively larger and of different color. This highlights the key features (i.e., the color and size of the head) to distinguishing weevil and vespula. Subsequently, the teacher provides another image of weevil which is visually very different from the query in terms the size of body and head, but is in the same category.

## 6 Conclusion

We introduced a novel interactive protocol for teaching an active leaner. We first derived a general performance guarantee for the greedy teacher with arbitrary active learners, and then provided a stronger performance bound for teaching GBS learners with a greedy teacher under mild assumptions. In addition, we also demonstrated the limitations of greedy teachers by showing that 1) the performance of the greedy teacher can be arbitrarily worse than that of the optimal teacher due to the constrained choices of contrastive examples; and 2) the greedy teaching strategy for the GBS learner is not guaranteed to be better than GBS alone. Lastly, we provided empirical results to demonstrate the effectiveness and limitations of the greedy teachers. We believe that our results have taken a significant step in bringing active learning and machine teaching closer to real-world settings in which the teacher collaborates with the learner to achieve the learning goal.

## 7 Social Impact

In this work, we provided an greedy teaching algorithm for active learners with theoretical guarantees. Our algorithm is derived for teaching an active leaner in a collaborative setting to achieve some learning goal, which can be potentially helpful for improving many real-world interactive learning systems, e.g., automated tutoring or multi-agent learning system.

However, just as a coin has two sides, our algorithm can be easily extended to poisoning the active learner by change the objective of the teacher. This can potentially threaten the safety of deploying machine learning models in online products.

## Acknowledgments and Disclosure of Funding

We thank Sandra Zilles for the helpful discussion and valuable feedback. This work is supported in part by NSF IIS-2040989 , and a C3.ai DTI Research Award 049755.

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
