# Appendix

## A  Datasets

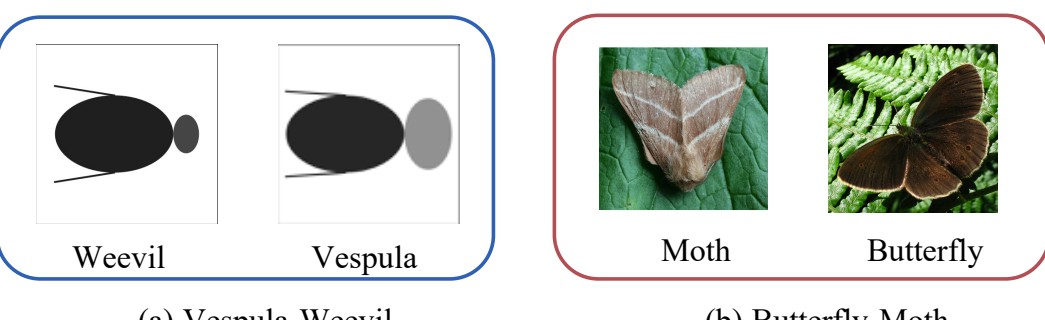

|  |  |
| :---: | :---: |
| (a) Vespula-Weevil | (b) Butterfly-Moth |

**Figure 1:** An visualization of the data in `Vespula-Weevil` and `Butterfly-Moth` datasets.

We visualize the samples from both `Vespula-Weevil` and `Butterfly-Moth` datasets in Figure 1. For `Vespula-Weevil` dataset, the images were generated by varying its body size and color, and also the head size and color. To distinguish each image, only a 2-d feature, i.e., 1) the ratio of the size of body and head; and 2) the contrast of the color of head and body, is sufficient. For `Butterfly-Moth` dataset, there are four species in it. Specifically, there are two different species of butterflies (Peacock butterfly and Ringlet butterfly) and also two different species of moths (Catepillar moth and Tiger moth). In general, it is much easier to classify Peacock butterfly and caterpillar moth. However, Tiger moth and Ringlet butterfly are hard to be correctly classified due to the visual similarity.

## B  Extension to General Costs

When the cost is non-uniform, i.e., $c(x) \neq c(x')$ if $x \neq x'$. Then, we can select the example at iteration $t$ based on the following rule

$$x_t^c \in \arg\max \frac{\Delta(x|x_{1:t-1}^c, q, \mathcal{H})}{c(x)}. \tag{1}$$

## C  Proofs of Theorem 1

**Definition 1** (**Pointwise submodularity ratio**). *For any sequence function $f$, the pointwise submodularity ratio with respect to any sequences $\sigma$, $\sigma'$ and query function $q$ and hypothesis class $H$ is defined as*

$$\rho_H(\sigma, \sigma', q) = \min_{x \in \mathcal{X}} \frac{\Delta(x|\sigma, q, H)}{\Delta(x|\sigma \oplus \sigma', q, H)}. \tag{2}$$

**Definition 2** (**Pointwise backward curvature**). *For any sequence function $f$, the pointwise backward curvature with respect to any sequences $\sigma$, $\sigma'$ and query function $q$ and hypothesis class $H$ is defined as*

$$\gamma_H(\sigma, \sigma', q) = 1 - \frac{f(\sigma' \oplus \sigma|q, H) - f(\sigma|q, H)}{f(\sigma'|q, H) - f(\emptyset|q, H)}. \tag{3}$$

**Lemma 1.** *For any $(\mathcal{X}, \mathcal{H})$ and active learner with query function $q$, we have $\mathtt{OPT}^T \leq \mathtt{OPT}^{T+AL} \leq 2 \cdot \mathtt{OPT}^T$, where $\mathtt{OPT}^T$ is the classic teaching complexity.*

*Proof.* Suppose the optimal teaching sequence corresponding to $\mathtt{OPT}^T$ is $(x_1^t, ..., x_{\mathtt{OPT}^T}^t)$ and the optimal teaching sequence corresponding to $\mathtt{OPT}^{T+AL}$ is $(x_1^q, x_1^c, ..., x_m^q, x_m^c)$ with $m = \mathtt{OPT}^{T+AL}/2$.

We prove the lemma by contrapositive. Suppose that $\mathtt{OPT}^{\mathtt{T+AL}} < \mathtt{OPT}^{\mathtt{T}}$. For this case, the teaching sequence $(x_1^t, ..., x_{\mathtt{OPT}^{\mathtt{T}}}^t)$ is not optimal. This contradicts with the definition of $\mathtt{OPT}^{\mathtt{T}}$. Therefore, we must have $\mathtt{OPT}^{\mathtt{T}} \leq \mathtt{OPT}^{\mathtt{T+AL}}$.

To show that $\mathtt{OPT}^{\mathtt{T+AL}} \leq 2 \cdot \mathtt{OPT}^{\mathtt{T}}$, consider the following case, we can replace $x_i^c$ with $x_i^t$ for all $i$. Then when $m = \mathtt{OPT}^{\mathtt{T}}$, the sequence $(x_1^q, x_1^c, ..., x_m^q, x_m^c)$ must cover all the incorrect hypotheses. Therefore, we can conclude that the optimal teaching sequence for $\mathtt{OPT}^{\mathtt{T+AL}}$ must be no longer than $2 \cdot \mathtt{OPT}^{\mathtt{T}}$.

$\square$

**Theorem 1.** *The sample complexity of the $\alpha$-approximate greedy algorithm for any active learner with a initial hypothesis class $\mathcal{H}$ is at most*

$$\left( \frac{\alpha \cdot \mathcal{O}\left(\log|\mathcal{H}| \cdot \log\left(|\mathcal{H}|/\gamma^g\right)\right)}{\rho^g \gamma^g \log(\gamma^g/(\gamma^g - 1))} + \frac{\alpha \cdot \mathcal{O}\left(\log\left(|\mathcal{H}|/\gamma^g\right)\right)}{\rho^g \gamma^g} \right) \cdot OPT^{T+AL}, \tag{4}$$

*where $\gamma^g = \max_{H \in \mathcal{H}'} \gamma_H^g$ and $\rho^g = \min_{H \in \mathcal{H}'} \rho_H^g$ with*

$$\gamma_H^g = \max_{i \geq 1} \gamma_H(\sigma^H, x_{1:i}^H, q), \quad \rho_H^g = \min_{i,j \geq 0} \rho_H(x_{1:i}^H, \sigma_{1:j}^H, q), \tag{5}$$

*which are computed with respect to the greedy teaching sequence $x^H$ and the optimal teaching sequence $\sigma^H$ for the corresponding active learner with a initial hypothesis class $H \in \mathcal{H}' = \{H^q(X) | X \in \mathcal{X}^\star \wedge |H^q(X)| \geq 2\}$ and query function $q$.*

*Proof.* Suppose that the optimal teaching sequence for active learner with a initial hypothesis class $\mathcal{H}$ is $\sigma$, and $K = \mathtt{OPT}^{\mathtt{T+AL}}$, we first have the following holds by using the definition of $\rho^g$ (we omitted the dependency on $q$ and $\mathcal{H}$ in $\Delta(\cdot)$ and $f(\cdot)$ for clarity)

$$\Delta(\sigma|x_{1:t}^c) = \sum_{i=1}^{K} \Delta(\sigma_i|x_{1:t}^c \oplus \sigma_{1:i-1}) \tag{6}$$

$$\leq \frac{1}{\rho^g} \cdot \sum_{i=1}^{K} \Delta(\sigma_i|x_{1:t}^c) \tag{7}$$

$$\leq \frac{K}{\rho^g} \cdot \max_{x \in \mathcal{X}} \Delta(x|x_{1:t}^c). \tag{8}$$

Then, using the definition of backward curvature ($f(\emptyset) = 0$ in our case), we get

$$\gamma^g \cdot f(x_{1:t}^c) \geq f(x_{1:t}^c) - f(x_{1:t}^c \oplus \sigma) + f(\sigma). \tag{9}$$

By combining the first results, we get

$$f(\sigma) - \gamma^g \cdot f(x_{1:t}^c) \leq \Delta(\sigma|x_{1:t}^c) \tag{10}$$

$$\leq \frac{K}{\rho^g} \cdot \max_{x \in \mathcal{X}} \Delta(x|x_{1:t}^c). \tag{11}$$

By the assumptions of $\alpha$-approximate greedy algorithm, the marginal gain of selected example $x_{t+1}^c$ by the greedy policy satisfies

$$\Delta(x_{t+1}^c|x_{1:t}^c) \geq \frac{1}{\alpha} \cdot \max_{x \in \mathcal{X}} \Delta(x|x_{1:t}^c). \tag{12}$$

This implies the objective at iteration $t + 1$ can be lower bounded by

$$\frac{1}{\alpha} \cdot \max_{x \in \mathcal{X}} \Delta(x|x_{1:t}^c) \geq \frac{\rho^g}{\alpha K} \cdot (f(\sigma) - \gamma^g \cdot f(x_{1:t}^c)) \tag{13}$$

$$\Rightarrow \quad f(x_{1:t}^c) + \Delta(x_{t+1}^c|x_{1:t}^c) \geq \frac{\rho^g}{\alpha K} \cdot f(\sigma) + \left(1 - \frac{\gamma^g \rho^g}{\alpha K}\right) \cdot f(x_{1:t}^c). \tag{14}$$

Then, we get the following recursive form of the greedy performance

$$f(x_{1:t+1}^c) \geq \frac{\rho^g}{\alpha K} \cdot f(\sigma) + \left(1 - \frac{\gamma^g \rho^g}{\alpha K}\right) \cdot f(x_{1:t}^c). \tag{15}$$

By solving the recursion, we have

$$f(x_{1:T_1}^c) \geq f(\sigma) \cdot \sum_{i=1}^{T_1-1} \frac{\rho^g}{\alpha K} \cdot \prod_{t=0}^{i-1} \left(1 - \frac{\rho^g \gamma^g}{\alpha K}\right)^t \tag{16}$$

$$= \frac{f(\sigma)}{\gamma^g} \cdot \left(1 - \left(1 - \frac{\rho^g \gamma^g}{\alpha K}\right)^{T_1}\right). \tag{17}$$

Since $(1+a)^x \leq e^{ax}$ when $x \geq 0$, we have

$$f(x_{1:T_1}^c) \geq \frac{f(\sigma)}{\gamma^g} \cdot \left(1 - e^{-(\rho^g \gamma^g \cdot T_1)/(\alpha K)}\right). \tag{18}$$

Then, plugging the following in equation 18,

$$T_1 = \frac{\alpha \cdot \mathcal{O}(\log(|\mathcal{H}|/\gamma^g))}{\rho^g \gamma^g} \cdot \text{OPT}^{\text{T+AL}}, \tag{19}$$

we get

$$f(x_{1:T_1}^c) \geq \frac{|\mathcal{H}|}{\gamma^g} \cdot (1 - \frac{1}{\mathcal{O}(|\mathcal{H}|)}) = \frac{|\mathcal{H}|}{\gamma^g}. \tag{20}$$

Therefore, the above results tells us that, if we run greedy teaching policy for $T_1$ steps, we are guaranteed to cover at least $|\mathcal{H}|/\gamma^g$ hypotheses. When $\gamma^g = 1$, we are done.

When $\gamma^g > 1$, we denote the remaining hypotheses as $\mathcal{H}_2$ (less than $(1 - 1/\gamma^g) \cdot |\mathcal{H}|$). Since $\mathcal{H}_2 \subseteq \mathcal{H}$, we must have the optimal teaching complexity smaller than $2 \cdot \text{OPT}^{\text{T+AL}}$ (by Lemma 1). Therefore, we can recursively apply the above bound. If we continue run the original greedy algorithm, we are guaranteed to cover $|\mathcal{H}_2|/\gamma^g$ in

$$T_2 \leq \frac{\alpha \cdot \mathcal{O}(\log(|\mathcal{H}_2|/\gamma^g))}{\rho^g \gamma^g} \cdot \text{OPT}^{\text{T+AL}} \tag{21}$$

$$= \frac{\alpha \cdot \mathcal{O}(\log(|\mathcal{H}| \cdot (\gamma^g - 1)/(\gamma^g)^2))}{\rho^g \gamma^g} \cdot \text{OPT}^{\text{T+AL}}. \tag{22}$$

We now can write it in terms of $T_n$,

$$T_n \leq \frac{\alpha \cdot \mathcal{O}(\log(|\mathcal{H}| \cdot (\gamma^g - 1)^{(n-1)}/(\gamma^g)^n))}{\rho^g \gamma^g} \cdot \text{OPT}^{\text{T+AL}}. \tag{23}$$

Therefore, the total complexity to cover all the hypotheses except the target hypothesis will be

$$T = \sum_{i=1}^n T_i \leq \sum_{i=1}^n \frac{\alpha \cdot \mathcal{O}(\log(|\mathcal{H}| \cdot (\gamma^g - 1)^{(i-1)}/(\gamma^g)^i))}{\rho^g \gamma^g} \cdot \text{OPT}^{\text{T+AL}}. \tag{24}$$

Since if there is only one hypothesis left, the algorithm will stop. Therefore, we must have the following upper bound for $n$

$$\left(1 - \frac{1}{\gamma^g}\right)^n \leq \frac{1}{|\mathcal{H}|} \quad \Rightarrow \quad n \leq \left\lceil \frac{\log |\mathcal{H}|}{\log(\gamma^g/(\gamma^g - 1))} \right\rceil. \tag{25}$$

Therefore, by plugging $n = \log |\mathcal{H}|/\log(\gamma^g/(\gamma^g - 1)) + 1$ in, we get the total complexity

$$T = \sum_{i=1}^n T_i \tag{26}$$

$$\leq \sum_{i=1}^n \frac{\alpha \cdot \mathcal{O}\left(\log((|\mathcal{H}|/\gamma^g) \cdot (\gamma^g - 1)^{(i-1)}/(\gamma^g)^{(i-1)})\right)}{\rho^g \gamma^g} \cdot \text{OPT}^{\text{T+AL}} \tag{27}$$

$$\leq \frac{n\alpha \cdot \mathcal{O}\left(\log\left((|\mathcal{H}|/\gamma^g) \cdot \sqrt{(\gamma^g - 1)^{(n-1)}/(\gamma^g)^{(n-1)}}\right)\right)}{\rho^g \gamma^g} \cdot \text{OPT}^{\text{T+AL}} \tag{28}$$

$$\leq \frac{n\alpha \cdot \mathcal{O}\left(\log(|\mathcal{H}|/\gamma^g)\right)}{\rho^g \gamma^g} \cdot \text{OPT}^{\text{T+AL}} \tag{29}$$

$$\leq \left(\frac{\alpha \cdot \mathcal{O}\left(\log |\mathcal{H}| \cdot \log\left(|\mathcal{H}|/\gamma^g\right)\right)}{\rho^g \gamma^g \log(\gamma^g/(\gamma^g - 1))} + \frac{\alpha \cdot \mathcal{O}\left(\log\left(|\mathcal{H}|/\gamma^g\right)\right)}{\rho^g \gamma^g}\right) \cdot \text{OPT}^{\text{T+AL}}. \tag{30}$$

□

# D   Proofs of Theorem 2

**Theorem 2.** *There exists constraint function $\xi$, version space $\mathcal{H}$ and active learner with query function $q$, such that the sample complexity of the greedy teacher is at least $\Omega(\sqrt{|\mathcal{H}|}) \cdot OPT^{T+AL}$.*

*Proof.* To see this, consider the following realizable example (please refer to Figure 2).

For each $x_i$, we have the following cases

- $x_1$: it can cut away $m/6$ hypotheses from the version space.
- $x_2$: it can cut away $m/6 + 1$ hypotheses from the version space.
- $x_3$: it can cut away $m/6 - 1$ hypotheses from the version space.
- $x_i$ ($i \geq 4$): it can cut away $\sqrt{m} + 4 - i$ hypotheses from the version space.

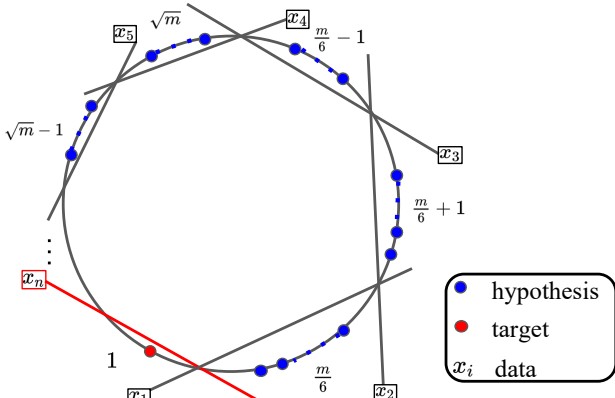

**Figure 2:** An illustrative example when the greedy teacher can be arbitrary worse than the optimal teacher. We visualize the data points and hypotheses in the dual space, where each hypothesis is a point and each data point is a hyperplane. Specifically, we use ● denotes the hypotheses that are not the target one, ● denotes the target hypothesis to teach, and $x_i$ are the data points.

Obviously, $n = \sqrt{m} + 3$. There are in total $\sqrt{m} + 3$ examples in the data set. The total number of hypotheses are $|\mathcal{H}| = m$. We assume that $m = (6k + 6)^2$ where $k \in \mathbb{N}_+$, which means $\sqrt{m} < m/6 - 1$. In addition, we have the following assumptions on the active learner and the constraint $\xi$:

- Active learner: we consider the GBS learner.
- Constraint $\xi$: given the query, the teacher can only choose contrastive examples from the immediate neighbors of the query point. For example, if the learner picks $x_4$, then the teacher can only pick contrastive examples from the constrained set $\{x_3, x_5\}$.

**Optimal teacher without constraints**: for the optimal teacher without the constraints, the total sample complexity is 2. That is, no matter what the active learner chooses, the teacher can always choose $x_n$, which cover all the hypotheses except for $h^\star$. This gives us $\text{OPT}^{\text{T+AL}} = \mathcal{O}(1)$.

**Optimal teacher with constraints**: for the optimal teacher with the constraints, the total sample complexity is 4. That is, the active learner first queries $x_2$, since it cuts the version space most evenly (so, it is most favored by the GBS learner). Then, the teacher will select $x_4$. In the next iteration, the learner will query $x_1$, and the techer will pick $x_n$, which cut away all the hypotheses except the target hypothesis.

**Greedy teacher with constraints**: for the greedy teacher with the constraints, the total complexity is $\sqrt{m} + 3$. At the first iteration, the learner queries $x_2$, and the greedy teacher will select $x_1$. Then, the learner will query $x_3$, and the teacher picks $x_4$. As we continue, for iteration $i \geq 2$, the learner will query $x_{i+1}$, and correspondingly, the teacher will pick $x_{i+1}$. This process will end until $x_n$ given to the learner. Therefore, all the examples will be needed to teach the learner with greedy teacher.

In the above example, we have the sample complexity for the greedy teacher at least

$$\Omega(\sqrt{m}) \cdot \text{OPT}^{\text{T+AL}} = \Omega(\sqrt{|\mathcal{H}|}) \cdot \text{OPT}^{\text{T+AL}}. \tag{31}$$

However, the bound in Theorem 1 also depends on $\rho^g$ and $\gamma^g$. In this example, the length of the optimal sequence must be 1 (see the reasoning in optimal teacher without constraints), because the example $x_n$ is sufficient to teach the learner the target hypothesis. Therefore, we can get

$$\rho^g = 1 \quad \text{and} \quad \gamma^g = 1. \tag{32}$$

Since when $\gamma^g \to 1$, the L.H.S. in the parentheses of equation 4 in Theorem 1 is 0. Therefore, the entire bound will be simplified to

$$\alpha \cdot \mathcal{O}\left(\log\left(|\mathcal{H}|\right)\right) \cdot \text{OPT}^{\text{T+AL}}. \tag{33}$$

To compute $\alpha$ for the example, we can directly use the definition

$$\alpha = \max_t \frac{\max_{x \in \mathcal{X}} \Delta(x|x_{1:t-1}^c, q)}{\max_{x \in \xi(x_t^q)} \Delta(x|x_{1:t-1}^c, q)} \approx \sqrt{m}. \tag{34}$$

Since $\log|\mathcal{H}| = o(\sqrt{m})$, and combining the above reasoning, we can conclude that the linear dependency on $\alpha$ cannot be avoided in the bound for general cases.

$\square$

# E    Proof of Theorem 3 and Remark 2

**Definition 3** ($k$-**neighborly**). *Consider the graph $(V, E)$ with vertex set $V = \mathcal{X}$, and edge set $E = \{(x, x')|d_{\mathcal{H}}(x, x') \le k, \ \forall \ x, x' \in \mathcal{X}\}$, where $d_{\mathcal{H}}(x, x') = |\{h|h \in \mathcal{H} \text{ and } h(x) \ne h(x')\}|$. The query and hypotheses space $(\mathcal{X}, \mathcal{H})$ is k-neighborly if the induced graph is connected.*

**Definition 4** (**Coherence parameter**). *The coherence parameter for $(\mathcal{X}, \mathcal{H})$ is defined as*

$$c^\star(\mathcal{X}, \mathcal{H}) := \min_P \max_{h \in \mathcal{H}} \left| \sum_{\mathcal{X}} h(x) dP(x) \right|, \tag{35}$$

*where we minimize over all possible distribution on $\mathcal{X}$.*

**Lemma 2.** *[Nowak, 2008] Assume that $(\mathcal{X}, \mathcal{H})$ is k-neighborly, and the coherence parameter is $c^\star$. Then, for every $\mathcal{H}' \subset \mathcal{H}$, the query $x$ selected according to GBS must reduce the viable hypotheses by a factor of at least $(1 + c^\star)/2$, i.e.,*

$$\left| \sum_{h \in \mathcal{H}'} h(x) \right| \le c^\star \cdot |\mathcal{H}'|, \tag{36}$$

*or the set $\mathcal{H}'$ is small, i.e.,*

$$|\mathcal{H}'| \le \frac{k}{c^\star}. \tag{37}$$

*Proof.* For each $\mathcal{H}' \subset \mathcal{H}$, consider the following two situations,

1. $\min_{x \in \mathcal{X}} |\mathcal{H}'|^{-1} |\sum_{h \in \mathcal{H}'} h(x)| \le c^\star$,

2. $\min_{x \in \mathcal{X}} |\mathcal{H}'|^{-1} |\sum_{h \in \mathcal{H}'} h(x)| > c^\star$.

For the first situation where $\min_{x \in \mathcal{X}} |\mathcal{H}'|^{-1} |\sum_{h \in \mathcal{H}'} h(x)| \le c^\star$, GBS will query the corresponding $x$ that minimize $|\mathcal{H}'|^{-1} |\sum_{h \in \mathcal{H}'} h(x)|$. This ensures that $\left| \sum_{h \in \mathcal{H}'} h(x) \right| \le c^\star \cdot |\mathcal{H}'|$.

For the second situation where $\min_{x \in \mathcal{X}} |\mathcal{H}'|^{-1} |\sum_{h \in \mathcal{H}'} h(x)| > c^\star$, we claim that there must exists $x^+, x^- \in \mathcal{X}$ such that $|\mathcal{H}'|^{-1} \sum_{h \in \mathcal{H}'} h(x^+) \ge c^\star$ and $|\mathcal{H}'|^{-1} \sum_{h \in \mathcal{H}'} h(x^-) < -c^\star$. To see this, recalling that

$$c^\star = c^\star(\mathcal{X}, \mathcal{H}) = \min_P \max_{h \in \mathcal{H}} \left| \sum_{\mathcal{X}} h(x) dP(x) \right|. \tag{38}$$

Then, we must have the following hold

$$c^\star \geq |\mathcal{H}'|^{-1} \left| \sum_{h \in \mathcal{H}'} \sum_{\mathcal{X}} h(x) dP(x) \right|, \tag{39}$$

where $P$ is the corresponding minimizer in equation 38. If

$$|\mathcal{H}'|^{-1} \sum_{h \in \mathcal{H}'} h(x^+) \geq c^\star, \ \forall x \in \mathcal{X} \quad \text{or} \quad |\mathcal{H}'|^{-1} \sum_{h \in \mathcal{H}'} h(x^+) < -c^\star, \ \forall x \in \mathcal{X}, \tag{40}$$

then equation 39 won't be satisfied, which leads to a contradiction. Therefore, we proved the claim.

Since $(\mathcal{X}, \mathcal{H})$ is $k-$neighborly, there exists a sequence of examples connecting $x^+$ and $x^-$. In the sequence, every two immediate neighbors are $k-$neighborhood (i.e., at most $k$ hypotheses in $\mathcal{H}$ predicts differently on them). Besides, there also must exists two neighbors, say $x, x'$, in the sequence such that the signs of $|\mathcal{H}'|^{-1} \sum_{h \in \mathcal{H}'} h(x)$ and $|\mathcal{H}'|^{-1} \sum_{h \in \mathcal{H}'} h(x')$ are different. Without the loss of generality, let's assume that $|\mathcal{H}'|^{-1} \sum_{h \in \mathcal{H}'} h(x) > c^\star$ and $|\mathcal{H}'|^{-1} \sum_{h \in \mathcal{H}'} h(x') < -c^\star$. Following the above observation, we immediately have the following two inequalities,

$$\sum_{h \in \mathcal{H}'} h(x) - \sum_{h \in \mathcal{H}'} h(x') \geq 2c^\star |\mathcal{H}'|, \tag{41}$$

$$\left| \sum_{h \in \mathcal{H}'} h(x) - \sum_{h \in \mathcal{H}'} h(x') \right| \leq 2k, \tag{42}$$

where the second inequality follows from the $k-$neighborly condition. By combining these two inequalities, we get the desired results

$$|\mathcal{H}'| < \frac{k}{c^\star}. \tag{43}$$

$\square$

**Lemma 3.** *For GBS learner (i.e., $\beta = 1$), if $(\mathcal{X}, \mathcal{H})$ is k-neighborly and with coherence parameter $c^\star$, then $\rho^g \geq \min\{(1 - c^\star)/(1 + c^\star), c^\star/(k - c^\star)\}$*

*Proof.* Recall the definition of $\rho$ (we omitted the dependency on $H$ for clarity),

$$\rho(\sigma, \sigma', q) = \min_{x \in \mathcal{X}} \frac{\Delta(x|\sigma, q)}{\Delta(x|\sigma \oplus \sigma', q)}. \tag{44}$$

The marginal gain is the sum of the marginal gain of the learner's query and that of the contrastive example $x$. Given the history $\sigma$ (i.e., the teaching sequence), let's denote the marginal gain of learner's query as $\Delta^q(\sigma)$, and the marginal gain of the contrastive example as $\Delta^c(x|\sigma)$ (exclude the gain overlapped with that of the learner's query). Then, we can rewrite $\rho$ as the following,

$$\rho(\sigma, \sigma', q) = \min_{x \in \mathcal{X}} \frac{\Delta^q(\sigma) + \Delta^c(x|\sigma)}{\Delta^q(\sigma \oplus \sigma') + \Delta^c(x|\sigma \oplus \sigma')} \tag{45}$$

$$\geq \frac{\Delta^q(\sigma)}{\Delta^q(\sigma \oplus \sigma')}. \tag{46}$$

Since $(\mathcal{X}, \mathcal{H})$ is $k$-neighborly, and the coherence parameter is $c^\star$, therefore, applying Lemma 2, we must have

$$\Delta^q(\sigma) \geq \left( \frac{1 - c^\star}{2} \right) \cdot |H^q(\sigma)|, \tag{47}$$

if $|H^q(\sigma)| > k/c^\star$. This further implies that

$$\Delta^q(\sigma \oplus \sigma') \leq \left( \frac{1 + c^\star}{2} \right) \cdot |H^q(\sigma)|, \tag{48}$$

By combining these two, we can get (when $|H^q(\sigma)| > k/c^\star$)

$$\rho(\sigma, \sigma', q) \geq \frac{1 - c^\star}{1 + c^\star}. \tag{49}$$

For the case of $|H^q(\sigma)| \leq k/c^\star$, we simply have the following

$$\rho(\sigma, \sigma', q) \geq \frac{1}{k/c^\star - 1} = \frac{c^\star}{k - c^\star}. \tag{50}$$

Therefore, we can conclude

$$\rho^g \geq \min\left\{\frac{1 - c^\star}{1 + c^\star}, \frac{c^\star}{k - c^\star}\right\}. \tag{51}$$

$\square$

**Theorem 3.** *For the GBS learner with a initial hypothesis class $\mathcal{H}$ and ground set $\mathcal{X}$, if $(\mathcal{X}, \mathcal{H})$ is $k$-neighborly and with coherence parameter $c^\star$, then the sample complexity of the greedy teaching algorithm with any constraint function $\xi$ is at most*

$$\frac{\alpha}{\epsilon} \cdot \mathcal{O}\left(\log^2(|\mathcal{H}|)\right) \cdot OPT^{T+AL}, \tag{52}$$

*where $\epsilon = \min\left\{(1 - c^\star)/(1 + c^\star), c^\star/(k - c^\star)\right\}$, and $\alpha \leq \max\{k/c^\star, 2/(1 - c^\star)\}$.*

*Proof.* By Theorem 1, we have that for any active learner and constraint function, the $\alpha$-approximate greedy teaching requires at most

$$\left(\frac{\alpha \cdot \mathcal{O}\left(\log|\mathcal{H}| \cdot \log\left(|\mathcal{H}|/\gamma^g\right)\right)}{\rho^g \gamma^g \log(\gamma^g/(\gamma^g - 1))} + \frac{\alpha \cdot \mathcal{O}\left(\log\left(|\mathcal{H}|/\gamma^g\right)\right)}{\rho^g \gamma^g}\right) \cdot OPT^{T+AL}. \tag{53}$$

Since $\gamma^g \geq 1$, then we have

$$\log(|\mathcal{H}|/\gamma^g) \leq \log(|\mathcal{H}|). \tag{54}$$

Now, consider the following function with $x > 1$

$$g(x) = \frac{a}{x \log(x/(x - 1))} + \frac{1}{x}, \tag{55}$$

of which the derivative is

$$\frac{\mathrm{d}}{\mathrm{d}x} g(x) = \frac{a - (x - 1) \cdot \log^2(x/(x - 1)) - (ax - a) \cdot \log(x/(x - 1))}{(x - 1) \cdot x^2 \cdot \log^2(x/(x - 1))}. \tag{56}$$

It's easy to verify that when $a \geq 2$, the derivative must be non-negative. Therefore, the function $g(x)$ is monotonically increasing. By replacing $a$ with $\mathcal{O}(\log(|\mathcal{H}|))$[2] and $x$ with $\gamma^g$, we have (since $\gamma^g < |\mathcal{H}|$)

$$\frac{\mathcal{O}(\log(|\mathcal{H}|))}{\gamma^g \log(\gamma^g/(\gamma^g - 1))} + \frac{1}{\gamma^g} \leq \frac{\mathcal{O}(\log(|\mathcal{H}|))}{|\mathcal{H}| \log(|\mathcal{H}|/(|\mathcal{H}| - 1))} + \frac{1}{|\mathcal{H}|} \leq \mathcal{O}(\log(|\mathcal{H}|)). \tag{57}$$

By plugging in the above results to equation 53, we simplifiy it to

$$\frac{\alpha}{\rho^g} \cdot \mathcal{O}(\log^2(|\mathcal{H}|)) \cdot OPT^{T+AL}. \tag{58}$$

By Lemma 3, we can finally conclude that for GBS learner, with $k$-neighborly $(\mathcal{X}, \mathcal{H})$ and coherence parameter $c^\star$, the sample complexity for $\alpha$-approximate greedy teaching policy is at most

$$\frac{\alpha}{\epsilon} \cdot \mathcal{O}(\log^2(|\mathcal{H}|)) \cdot OPT^{T+AL}, \quad \text{where} \quad \epsilon = \min\left\{\frac{1 - c^\star}{1 + c^\star}, \frac{c^\star}{k - c^\star}\right\}. \tag{59}$$

---

[2]Without the loss of generality, we can assume $|\mathcal{H}| \geq 8 > e^2$

To further bound the term $\alpha$, recall the definition of $\alpha$,

$$\alpha = \max_t \frac{\max_{x \in \mathcal{X}} \Delta(x | x^c_{1:t-1}, q)}{\max_{x \in \xi(x^q_t)} \Delta(x | x^c_{1:t-1}, q)}. \tag{60}$$

Since the learner's query is the same, the only affecting term is the marginal gain of the contrastive example provided by the teacher. Consider the extreme case where the numerator covers the entire version space but the all the examples in $\xi(x^q_t)$ cannot cover any hypotheses in the version. This is equivalent to say that the contribution of $x$ in the denominator is $0$. Then, by Lemma 2, when $|H^q(x^c_{1:t-1})| > k/c^\star$, we must have the following,

$$\alpha \leq \frac{|H^q(x^c_{1:t-1})|}{(1-c^\star)/2 \cdot |H^q(x^c_{1:t-1})|} = \frac{2}{1-c^\star}. \tag{61}$$

Otherwise, when $|H^q(x^c_{1:t-1})| \leq k/c^\star$, we must have

$$\alpha \leq \frac{k/c^\star}{1} = \frac{k}{c^\star}. \tag{62}$$

By combining them, we can conclude that

$$\alpha \leq \max\left\{ \frac{2}{1-c^\star}, \frac{k}{c^\star} \right\}. \tag{63}$$

$\square$

**Remark 2.** *The sample complexity of GBS with greedy teacher (even without the constraints) is not guaranteed to be smaller than that of GBS alone.*

*Proof.* To show this, we provide a realizable example below. In Figure 3, each circle is corresponding to a data point, and each dot is a hypothesis in the version space. Specifically, the blue $\bullet$ are the incorrect hypotheses and the red $\bullet$ is the correct/target hypothesis. For each data point $x_i$, all the hypotheses covered by the corresponding circle are those hypotheses that predict incorrectly on $x_i$. Therefore, upon the learner receives the data point $x_i$, all the hypotheses covered by the corresponding circle will be immediately removed from the learner's version space.

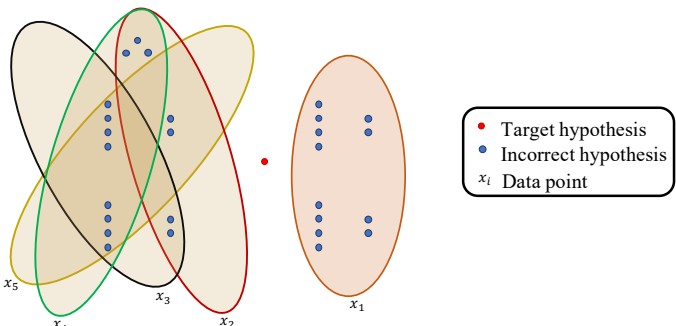

**Figure 3:** An illustrative example when GBS alone can achieve better sample complexity than that of GBS with greedy teacher.

- For GBS learner alone, it first selects the query $x_1$ which covers/removes the right part of the hypotheses. Then, it selects $x_2$, because $x_2$ is the most uncertain point among $\{x_2, x_3, x_4, x_5\}$. Lastly, it selects any point from $\{x_3, x_4, x_5\}$, leaving only the target hypothesis (red $\bullet$) in the version space. Therefore, the sequence for the GBS alone is $\{x_1, x_2, x_3\}$ or $\{x_1, x_2, x_4\}$ or $\{x_1, x_2, x_5\}$, of which the total cost is 3.

- For GBS with a greedy teacher (with any constraints on the contrastive example), the GBS learner will first query $x_1$ and the teacher will select $x_4$ as the contrastive example. In the next round, the learner will query either $x_3$ (or $x_5$), and the teacher will select either $x_2$ or $x_5$ (or either $x_2$ or $x_3$). Then, only the target hypothesis is left. Therefore, the sequence for GBS with greedy teacher is $(x_1, x_4, x_3, x_2)$ or $(x_1, x_4, x_3, x_5)$ or $(x_1, x_4, x_5, x_2)$ or $(x_1, x_4, x_5, x_3)$, of which the cost is 4.

We can see that for GBS learner alone, it only requires $3$ examples to identify the target hypothesis, whereas for GBS with an unconstrained teacher, it needs $4$ examples for identifying the target hypothesis. This example demonstrates that the greedy teacher is not always helpful for active learners, and sometimes it will even hurt the performance of the active learner.

$\square$