# OpenReview forum: "Teaching an Active Learner with Contrastive Examples"
_NeurIPS.cc/2021/Conference — NeurIPS 2021 Poster_

### Official Review · Reviewer_j9HW · 2021-07-08

**Rating:** 6
**Confidence:** 4

**Summary:**

The paper addresses the problem of teaching an active learning with an explanatory teacher that provides contrastive examples together with the query label. They show guarantees for the greedy version with respect to GBS (Nowak 2008) and provide 2 small data sets for the validation of the advantage of the greedy teacher and demonstrating the tradeoff between the level of greed and the constraint on the search space.

**Limitations And Societal Impact:**


yes

**Main Review:**

The problem setting addressed is interesting and novel - teaching an active learner with contrastive examples seems to be novel to my best knowledge. The greedy learner is not a new format, and I would expect the authors to better address that on the related work section.

The submission seems technically sound, and the claims seems to be well supported although I did not go through the proofs in detail. The experimental results capture an interesting tradeoff between the different constraints and show the greedy-teacher advantage. However more experimentation and even with some benchmark data could be more convincing.

There are still fundamental questions to be answered: in all the hypothesis reduction approaches the practical aspect is somewhat problematic in my view: for example: in Alg 2 we receive as output the target hypothesis? Can the authors bridge this setting to the practical use of their techniques? Namely, can we practically use such a techniques on real problems? How did they use these techniques on their small data sets in the experiments section? Can we approximate some target that will be of interest but can be a-priori defined on some validation set?
Additionally, there are still some details missing here: what is \Psi and how is it formally defined? Is it related to \xi in equation (4)?


Overall, the paper is clear but some parts need better clarity in particular in the experiments section see above.
Also why is there an upper script c on equation (2) on each x? is this upper script related to contrastiveness?
I think that reporoducing the results may be hard in this case as some parts are not clear (e.g. \xi, \psi).


**Time Spent Reviewing:**

8

---

> ### Author Response · Authors · 2021-08-05
> **Response to Reviewer j9HW**
>
> Thank you for carefully reviewing our paper! We appreciate that you think our problem setting is interesting and novel. Our response to your comments can be referred to below.
>
> ---
>
> **1. Practical aspect of hypothesis reduction approaches**
>
> - We agree that hypothesis reduction approaches have limited applicability in practice.
> “Version space” (hypothesis) -reduction approaches are useful tools for studying the theory of active learning [1,2,3,4,5] and machine teaching [6,7,8,9]. The notion of “version space”-based active learner has a long history in active learning---interestingly---as a practical alternative [1] to the classical information-theoretic (e.g., entropy reduction/uncertainty sampling) and decision-theoretic heuristics (e.g., value of information); the latter two being two popular heuristics for many practical active learning systems. It is also known that for the optimal decision tree problem---a canonical active learning problem---version space reduction and the entropy reduction heuristics are equivalent [2]. Hence we respectfully argue that version space reduction-based approaches have great potential in practical applications.
>
> - Although our work has a theoretical focus, we believe the technique and high-level ideas in our paper can be extended to real problems, but the effort should be spent on designing the notion for measuring the “hypothesis reduction” and the acquisition function of the active learner. For real problems, the theoretical analysis will be difficult and sometimes infeasible, as it highly depends on the specific problem.
>
> **2. Details missing & clarity about experiments**
> - We will release our code and dataset once our paper is made public. Regarding your questions, $\Psi$ defines the search radius of the contrastive examples (see line 285). For example, when the constraint function is “examples that are close to the learner’s query but with different labels”, and when $\Psi=1$, all of those examples that satisfy the constraint will be in the feasible set of contrastive examples; whereas when $\Psi=0.1$, only the closest 10% of examples that satisfy the constraint will be considered.
>
> **3. More experimentation**
> - Thank you for the suggestion. We agree that more experimentation will make the paper stronger, but also would shift the focus of our work as a theory-oriented package. Our goal of including the experiment section is to demonstrate that the greedy teacher is helpful in practice and also better illustrate our theory.
>
> **4. More discussion on greedy leaner**
> - Thanks for the suggestion. We will include the discussion about greedy learners in related works.
>
> **5. Why there is an superscript $c$ on equation (2)**
> - We apologize for the confusion. The superscript $c$ is to denote that the example is selected by the teacher, i.e., the **c**ontrastive example. Similarly, the superscript $q$ is to denote that the example is a learner's **q**uery.
>
> ---
> We hope that our responses can address your concerns well. If there are any other concerns, please let us know! We will be happy to make further responses! We are looking forward to hearing back from you! Thank you again for the reviews!
>
> ---
> References:
>
> [1] Simon Tong and Daphne Koller. Support Vector Machine Active Learning with Applications to Text Classification, ICML, 1998.
>
> [2] Alice X. Zheng, Irina Rish, and Alina Beygelzimer. Efficient test selection in active diagnosis via entropy approximation. In UAI ’05, Proceedings of the 21st Conference in Uncertainty in Artificial Intelligence, 2005.
>
> [3] Dasgupta, S. (2004).  Analysis of a greedy active learning strategy.  InNIPS: Advances in NeuralInformation Processing Systems 17, pp. 337–344. MIT Press
>
> [4]  Nowak, R. (2009).   Noisy generalized binary search.   InNIPS: Advances in Neural InformationProcessing Systems 22, pp. 1366–1374.485
>
> [5] Golovin, Daniel, and Andreas Krause. “Adaptive submodularity: Theory and applications in active learning and stochastic optimization.” Journal of Artificial Intelligence Research 42 (2011): 427-486.
>
> [6] Xiaojin Zhu, Adish Singla, Sandra Zilles, and Anna N. Rafferty. An overview of machine teaching. CoRR, abs/1801.05927, 2018.
>
> [7] Sandra Zilles, Steffen Lange, Robert Holte, and Martin Zinkevich. Models of cooperative teaching and learning. JMLR, 12(Feb):349–384, 2011.
>
> [8] Farnam Mansouri, Yuxin Chen, Ara Vartanian, Xiaojin Zhu, Adish Singla. Preference-Based Batch and Sequential Teaching: Towards a Unified View of Models, Farnam Mansouri, Yuxin Chen, Ara Vartanian, Xiaojin Zhu, Adish Singla. NeurIPS, 2019.
>
> [9] Sally A Goldman and Michael J Kearns. On the complexity of teaching. Journal of Computer and System Sciences, 50(1):20–31, 1995.

---

### Official Review · Reviewer_NhnX · 2021-07-16

**Rating:** 6
**Confidence:** 4

**Summary:**

This paper studies a scenario combining active learning and teaching: An active learner cooperates with a teacher, which provides contrastive instances to the learner to accelerate learning. The paper is basically theoretical-oriented, the main contributions are:
>- Provide the approximation ratio of the greedy teaching algorithm;
>- Provide improved results for greedy active learners.

**Limitations And Societal Impact:**

They are adequately addressed.

**Main Review:**

Originally: Good

The paper studies a new scenario combining active learning and teaching, which raises interesting theoretical and algorithmic challenges.

Quality:  Proofs seem to be incorrect.
In Equation (9) of theorem 1, there is a term $\log(\gamma^g/\gamma^g-1)$. However，$\gamma^g \leq 1$ according to the definition. I have checked the proofs in the appendix, and find that Equation (19) and (20) in the proof of theorem 1 may be incorrect. I think the correct bound for Equation (9) of theorem 1 is $T \geq \frac{\alpha}{\rho^g\gamma^g}\log(1/(1-\gamma^g)) \cdot OPT$, which can be derived from the proof process proposed in the paper. While this makes the example in theorem 2 incorrect, since \gamma^g cannot be 1 in this bound, while the proof of theorem 2 relies on this.

Clarity: Good

Significance: If I am right, the above issue should be corrected.

-------------------------
After rebuttal:
The above issue has been clarified. I do think the paper makes an interesting attempt towards the T+AL problem. It is expected that in the updated version, the suggestions raised in the reviewer-author discussions would be addressed.





**Time Spent Reviewing:**

3 hours

---

> ### Author Response · Authors · 2021-08-05
> **Response to Reviewer NhnX**
>
> Thank you for carefully reviewing our paper. We are glad to see that you recognize the originality and clarity of the paper. Regarding the issues raised in the review, please see our responses below.
>
> -----
> **1. $\gamma^g \leq 1$ and equation (19)&(20)**
>
> - By definition (see equation (7)&(10)), $\gamma^g \geq 1$. Specifically, in equation (10), $\sigma$ is the optimal teaching sequence. This means $\sigma$ is the maximizer of the function $f(\cdot|q)$. Hence, we must have $f(\sigma’\oplus\sigma|q) \leq f(\sigma|q)$. In addition, since $f(\emptyset|q)=0$, we must also have $f(\sigma’|q)> f(\emptyset|q)$. By combining these two, we can conclude that $$\frac{f(\sigma’\oplus\sigma|q)-f(\sigma|q)}{f(\sigma’|q)-f(\emptyset|q)}\leq 0$$. Hence, we must have $\gamma^g \geq 1$.
>
> - To be noted, our bound in Theorem 1 (equation (9) in the main paper) holds for any $\gamma^g\geq 1$. For the case of $\gamma^g=1$, the bound in Theorem 1 will be simplified to
> $$\frac{\alpha\cdot\mathcal{O}(\log(|\mathcal{H}|))}{\rho^g}\cdot \text{OPT}^{\text{T+AL}}$$, as the left-hand side term in the parentheses of equation (9) will vanish when $\gamma^g\rightarrow 1$.
> This simplified bound is exactly equation (19) in the appendix if setting $\gamma^g=1$. (We summarized this result in Remark 1. )
> -----
>
> **2. The example in theorem 2 is incorrect**
>
> - Regarding the example in the proof of Theorem 2, we have $\gamma^g=1$ and $\rho^g=1$. Therefore, if we plug their values in the bound above, we can get the following bound,  $$\alpha\cdot\mathcal{O}(\log(|\mathcal{H}|))\cdot \text{OPT}^{\text{T+AL}}$$. Furthermore, since $\alpha=\sqrt{|\mathcal{H}|}$ and $\log|\mathcal{H}| = o(\sqrt{|\mathcal{H}|})$. Therefore, we can see that  $$\Omega(\sqrt{|\mathcal{H}|})\cdot\text{OPT}^{\text{T+AL}}$$ is a lower bound, and the dependency on $\alpha$ cannot be avoided.
>
> -----
> The postfix-montone assumption is a commonly adopted assumption for obtaining the theoretical guarantees for greedy algorithms; see [1]. From this perspective, we would like to highlight the novelty of Theorem 1, as it goes beyond the postfix-monotone assumption and works for $\gamma^g > 1$, i.e., the sequence function needs not to be postfix-monotone. We believe that this result is a useful contribution and can be of separate interest for people studying the theoretical aspects of sequence optimization and optimal control.
>
>
> We hope that our responses can address your concerns well, and are helpful for improving the ratings. If you have any other concerns, please let us know! We will be happy to make further responses! We are looking forward to hearing back from you! Thank you again for the reviews!
>
> -----
> Reference:
>
> [1] Zhang, Z., Chong, E. K., Pezeshki, A., & Moran, W. (2015). String submodular functions with curvature constraints. IEEE Transactions on Automatic Control, 61(3), 601-616.

---

> > ### Comment · Reviewer_NhnX · 2021-08-20
> > **further question**
> >
> > Thanks for the response.
> >
> >  "By definition (see equation (7)&(10)), $\gamma^g \geq 1$."
> >
> > - I am quite confused about this explanation since, in Line 188-189 of the paper, it writes " For postfix monotone sequence functions, the pointwise backward curvature is smaller than 1 (Zhang et al., 2015)."

---

> > > ### Author Response · Authors · 2021-08-20
> > > **Response to further question**
> > >
> > > Thanks a lot for your further question. We are more than happy to address it. We sincerely hope that our response can resolve your concerns. Any follow-up questions are welcome.
> > >
> > > **1. "For postfix monotone sequence functions, the pointwise backward curvature is smaller than 1 (Zhang et al., 2015)."**
> > >
> > > - We are sorry for the confusion. We first explain why **"For postfix monotone sequence functions, the pointwise backward curvature is smaller than 1 (Zhang et al., 2015)."**
> > >
> > >   The backward curvature $\gamma$ is defined (equation(7)) as a function of **any** two sequences $\sigma, \sigma'$,
> > > $$\gamma(\sigma, \sigma', q) = 1 - \frac{f(\sigma'\oplus\sigma|q) - f(\sigma|q)}{f(\sigma'|q) - f(\emptyset|q)}.$$
> > >
> > >   When the sequence function $f(\cdot|q)$ is **postfix-monotone**, for any two sequences $\sigma, \sigma'$, we will have
> > > $$f(\sigma'\oplus\sigma|q) \geq f(\sigma|q), $$
> > > which implies that
> > > $$\gamma(\sigma, \sigma', q) = 1 - \frac{f(\sigma'\oplus\sigma|q) - f(\sigma|q)}{f(\sigma'|q) - f(\emptyset|q)} \leq 1.$$
> > >
> > >   **Note**: when $\sigma$ is the maximizer of $f(\cdot|q)$, then $\gamma(\sigma, \sigma', q) = 1$.
> > >
> > >
> > > **2. By definition (see equation (7)&(10)) $\gamma^g \geq 1$.**
> > >
> > > - In our case, $\gamma^g$ is a special case of $\gamma$. Specifically, $\gamma^g$ is defined as a function of the **optimal sequence $\sigma$** and an arbitrary sequence $\sigma'$, rather than two arbitrary sequences. This is because we want to derive an approximation ratio of the teaching sequence $\sigma'$ with respect to the **optimal teaching sequence $\sigma$**.
> > >
> > >   Since $\sigma$ is the optimal sequence, we must have
> > > $$\sigma \in \text{argmax}_{s} f(s|q).$$
> > > Therefore, we also have the following hold by the optimality of $\sigma$,
> > > $$f(\sigma'\oplus|q) - f(\sigma|q) \leq 0.$$
> > >   Now, consider the following two cases:
> > >   - $f(\cdot|q)$ is postfix-monotone:
> > >     $$f(\sigma'\oplus|q) - f(\sigma|q) = 0 \Rightarrow \gamma^g = 1$$
> > >   - $f(\cdot|q)$ is **NOT** postfix-monotone:
> > >
> > >     In this case, we can only have $f(\sigma'\oplus|q) - f(\sigma|q) \leq 0.$ The equality does not always hold. Although $\sigma$ is the optimal sequence, its optimality is **not** guaranteed when we append a prefix sequence $\sigma'$ to $\sigma$, i.e., $\sigma'\oplus \sigma$.  This is because the prefix $\sigma'$ will change the dynamics of the learner.
> > >
> > >
> > > In summary, our work goes beyond the assumption of postfix-monotone. When the function $f(\cdot|q)$ is postfix-monotone, we have $\gamma^g = 1$. When the function is not postfix-monotone, we have $\gamma^g \geq 1$.
> > >
> > > Thanks again for your careful reviews! We will highlight the difference between $\gamma$ and $\gamma^g$ in our revision to avoid confusion.

---

> > > > ### Comment · Reviewer_NhnX · 2021-08-21
> > > > **further question**
> > > >
> > > > Thanks for the in-time reply. While my doubts remain. For the explanation:
> > > >
> > > > Therefore, we also have the following hold by the optimality of $\sigma$,
> > > > $$f(\sigma'\oplus\sigma|q) - f(\sigma|q) \leq 0$$.
> > > >
> > > > - If a teaching sequence is concatenated with another one as in $f(\sigma'\oplus\sigma|q)$ , it means that the number of instances to teach is increased. Therefore, the number of hypotheses removed from the version space (i.e. $f(\cdot|q)$, according to the definition in Line 166-167 in the paper) should be increased. Thus I think we must have $f(\sigma'\oplus\sigma|q) - f(\sigma|q) \geq 0$.

---

> > > > > ### Author Response · Authors · 2021-08-21
> > > > > **Further response**
> > > > >
> > > > > Thanks a lot for your prompt reply! We are happy to address your remaining doubts!
> > > > >
> > > > > **1. The number of instances to teach is increased. Therefore, the number of hypotheses removed from the version space should be increased.**
> > > > >
> > > > > - Here, $\sigma$ and $\sigma'$ are the teaching sequences (the examples provided by the teacher only), which **don't include the learner's queries**.
> > > > >
> > > > >   Suppose that $\sigma = (x_1^c, x_2^c, ..., x_k^c)$, then we have
> > > > > $$f(\sigma|q) = f(x_{1:k}^c|q) =\left|\mathcal{H}\setminus H^q(x_{1:k}^c)\right|.$$
> > > > > By the definition of  $ H^q(x_{1:k}^c)$ (equation (1)), we have
> > > > > $$f(\sigma|q) = \left|\bigcup_{i=1}^k S(x_i^q)\cup S(x^c_i)\right|,$$
> > > > > where $x^q_i$ is the learner's query at each iteration $i$. Therefore, the entire sequence (including the learner's queries and the teaching sequence $\sigma$) is
> > > > > $$s_\sigma = (x^c_1, x^q_1, ..., x^c_k, x^q_k).$$
> > > > >   Similarly, let's assume the entire sequence corresponding to $\sigma'$ is
> > > > >   $$s_\sigma' = (z^c_1, z^q_1, ..., z^c_l, z^q_l),$$
> > > > >   which gives us that
> > > > > $$f(\sigma'|q) = \left|\bigcup_{i=1}^l S(z_i^q)\cup S(z^c_i)\right|,$$
> > > > >   For the sequence $\sigma'\oplus\sigma$, the corresponding sequence (including the learner's queries and the teacher's examples) is **NOT** the concatenation of $s_\sigma'$ and $s_\sigma$, i.e.,
> > > > > $$(z^c_1, z^q_1, ..., z^c_l, z^q_l)\oplus (x^c_1, {\color{red}x^q_1}, x^c_2, {\color{red}x^q_2}, ..., x^c_k, {\color{red}x^q_k}).$$
> > > > > This is because the learner's query depends on the state at each iteration. Therefore, when we concatenate $\sigma'$ before $\sigma$, it will change the learner's states and hence the queries (i.e., $(x_1^q, ..., x^q_k)$, highlighted in red in the above equation) corresponding to $\sigma$. This may make the teaching sequence $\sigma'\oplus\sigma$ no longer optimal, because the learner's queries will be changed.
> > > > >
> > > > >   **An illustrative example**: Consider the following constructed example for illustration. Suppose that there are in total $n$ hypothesis, and only one is the correct hypothesis. Let's further assume that the sequence correspond to $\sigma$ (the optimal teaching sequence) is
> > > > > $$s_\sigma = (x^c_1, x^q_1),$$
> > > > > and the sequence correspond to $\sigma'$ is
> > > > > $$s_\sigma' = (z^c_1, z^q_1).$$
> > > > > Suppose $x^c_1$ covers $\frac{n}{3}$ hypothesis, $x^q_1$ covers $\frac{2n}{3}-1$ hypothesis (disjoint with the coverage of $x^c_1$), $z^c_1$ covers $1$ hypothesis and $z^q_1$ covers $1$ hypothesis. Assume that the entire sequence correspond to $\sigma'\oplus \sigma$ is
> > > > > $$(z^c_1, z^q_1, x^c_1, {\color{red}y^q_1}),$$
> > > > > where $y^q_1 \neq x^q_1$. This is because the learner's state with sequence $(z^c_1, z^q_1, x^c_1)$is different with the learner's state with sequence $(x^c_1)$. Hence, the learner's query will also change correspondingly. Since in our theorem 1, we don't have any restrictions on the learner, then ${\color{red} y^q_1}$ can be arbitrary. Let's assume $y^q_1$ can only cover $1$ hypothesis. Therefore, we will have
> > > > > $$f(\sigma'\oplus\sigma|q) = S(z^c_1)\cup S(z^q_1) \cup  S(x^c_1) \cup  S(y^q_1) \leq 1+ 1+ \frac{n}{3} + 1 = \frac{n}{3} + 3,$$
> > > > > whereas
> > > > > $$f(\sigma|q) = S(x^c_1) \cup  S(x^q_1) = \frac{n}{3} + \frac{2n}{3} - 1 = n - 1.$$
> > > > > When $n > 9$, we must have
> > > > > $$n-1 > \frac{n}{3} + 3 \Rightarrow f(\sigma|q)  >f(\sigma'\oplus\sigma|q).$$
> > > > >
> > > > >
> > > > > We sincerely hope that our response can resolve your doubts. Any follow-up questions are welcome!

---

> > > > > > ### Comment · Reviewer_NhnX · 2021-08-21
> > > > > > **discussions**
> > > > > >
> > > > > > Many thanks for the explanations! I think they indeed address my previous doubt over $\gamma^g$. I suggest:
> > > > > > - putting the above explanations in the paper, or in the appendix.
> > > > > > - changing the r.h.s. of equation (7) to $1+ \frac{f(\sigma|q) - f(\sigma'\oplus\sigma|q)}{f(\sigma'|q) - f(\phi|q)}$.
> > > > > > - In the proof of theorem 1, $K$ is not explicitly defined. According to equation (6) in the appendix, $K$ seems to be the length of $\sigma$. I suggest providing an explicit definition.
> > > > > >
> > > > > > I think it is clearer for me to understand the results: The paper introduces a greedy teacher that teaches to shrink the version space aggressively. Without the active learner, this teacher would behave like the normal teacher in previous teaching algorithms. While if it is combined with the active learner, theorem 1 tells us the following things:
> > > > > > - If the active learner's query strategy does not affect the teacher a lot, i.e. $\gamma^g =1$, then the teaching cost will be dominated by the teacher.
> > > > > > - If the active learner's query strategy has significant influences on the teacher, i.e. $\gamma^g > 1$, then the teaching cost would be significantly increased.
> > > > > >
> > > > > > Even though I think the above results are reasonable, I still think the results in the current paper are limited:
> > > > > > - What I expected is a teaching algorithm that really "strengthens" the active learner, but the greedy teacher in the paper is just effective by only itself, or even is possible to be weakened by the active learner.
> > > > > > - I think one of the really interesting things shown by the paper is that there indeed exist some bad active learners that are not compatible with teaching. While the paper does not provide real examples of such learners.
> > > > > >
> > > > > > For the above concerns, I think the results in the paper are not that strong to meet my expectations. I would raise my score from 4 to 5. I will be happy to see that more practical algorithms, stronger theoretical results, and real examples of bad active learners provided in the paper.

---

> > > > > > > ### Comment · Reviewer_NhnX · 2021-08-21
> > > > > > > **further explanations**
> > > > > > >
> > > > > > > I think at the starting point of studying to combine active learning and teaching, it is necessary to answer:
> > > > > > > - is it always possible to make the teacher and the learner cooperate?
> > > > > > > - if not, under which situations it is impossible/possible?
> > > > > > > - does there exist effective algorithms?
> > > > > > >
> > > > > > > I think the current results in the paper only hide the above questions in the "OPT" term. In my view, analyzing this OPT solution is more important than proposing an approximation algorithm for it.

---

> > > > > > > ### Author Response · Authors · 2021-08-22
> > > > > > > **Response**
> > > > > > >
> > > > > > > Thank you very much for your valuable suggestions and constructive feedback! We really appreciate it! We are more than happy to address your raised concerns. We are open to any follow-up discussions.
> > > > > > >
> > > > > > > We would like to first highlight the main contributions of our paper:
> > > > > > > - A novel learner-teacher interaction framework.
> > > > > > > - A Upper bounds on the sample complexity of greedy teachers (Theorem 1).
> > > > > > > - Lower bounds on the sample complexity of greedy teachers (Theorem 2).
> > > > > > > - Upper bounds on the sample complexity of greedy teacher with GSB learner. (Theorem 3).
> > > > > > > - Negative results: Limitation of greedy learner (Remark 2)
> > > > > > > - Proof-of-concept experiments.
> > > > > > >
> > > > > > >
> > > > > > > **1. $\gamma^g=1 \Leftrightarrow$ the active learner's query strategy does not affect the teacher a lot; $\gamma^g>1 \Leftrightarrow$ the active learner's query strategy significant**
> > > > > > >
> > > > > > > - We respectfully disagree. $\gamma^g$ measures the degree of diminishing returns of the difference between marginal gains. No matter $\gamma^g > 1$ or $\gamma^g = 1$, the learner can always affect the teacher significantly, as the teacher's example depends on the learner's query (see Line 5 in Algorithm 2) and the constraint function $\xi(\cdot)$. More precisely, the parameter $\alpha$ is the quantity for measuring if the learner affects the teacher significantly or not.
> > > > > > >
> > > > > > > - To interpret $\gamma^g$, we believe it's more proper to say a larger $\gamma^g$ implies the learner is more adversary, i.e., non-cooperative, and hence the total cost will be increased. However, the cost cannot be increased significantly thanks to the teacher. This is reflected by our Remark 1. In Remark 1, when $\gamma^g \rightarrow |\mathcal{H}|-1$, the left-hand side term in the parentheses of equation (9) will dominate. Though for large $\gamma^g$, the cost will increase, the increased ratio is at most $O(\log(|\mathcal{H}|))$ compared to the original cost. However, for the active learner alone, due to its adversary nature, it may even select useless examples until it identifies the target hypothesis. This will incur a cost of $O(|\mathcal{H}|)$.
> > > > > > >
> > > > > > >
> > > > > > > **2. What I expected is a teaching algorithm that really "strengthens" the active learner, but the greedy teacher in the paper is just effective by only itself or even is possible to be weakened by the active learner.**
> > > > > > > - Thanks for the great suggestion! This is the ultimate goal of our research in teaching an active learner. The scope of our work is not to propose a new algorithm for a new problem. Instead, our work aims to provide a novel learner-teacher interaction framework and analyze the theoretical performance of greedy teachers under this setting. The greedy teaching algorithm is the most common algorithm in machine teaching, and it performs very well in teaching *passive* learners. However, surprisingly, our work demonstrated that the greedy teacher is not satisfactory when the learner is *active*. Though this is a negative result on the greedy teacher, we argue that the negative results are also part of the research progress. For example, the work by Sanjoy Dagupta (2004) [1] analyzed the theoretical performance of greedy active learners. He proved that the worst-case bound on the label complexity of greedy active learner is $O(|\mathcal{H}|)$. This was a negative result about greedy active learners. However, built upon his negative result, it has motivated a lot of future works by either proposing better active learning algorithms or better problem structures [2,3,4].
> > > > > > >
> > > > > > > - In addition, although the greedy teacher is not always guaranteed to be helpful, we empirical found that the greedy teacher is helpful in most cases.
> > > > > > >
> > > > > > >
> > > > > > > **3. I think one of the really interesting things shown by the paper is that there indeed exist some bad active learners that are not compatible with teaching. While the paper does not provide real examples of such learners.**
> > > > > > > - In general, those bad active learners can be constructed in an adversary way, which makes them meaningless. More importantly, such active learners may not be considered in practice. Most of the active learners, e.g., GBS learners, do not fall in this category.
> > > > > > >
> > > > > > >
> > > > > > > **4. In my view, analyzing this OPT solution is more important than proposing an approximation algorithm for it.**
> > > > > > > - If we want to analyze $\text{OPT}$ solution, we will need to be able to compute them first. Unfortunately, computing $\text{OPT}^{\text{T+AL}}$ (the length of the optimal sequence) is NP-hard, and computing the $\text{OPT}$ solution is even harder.
> > > > > > >
> > > > > > >
> > > > > > > Thanks again for your time. We enjoyed the discussion with you! Any follow-up discussions are welcome!
> > > > > > >
> > > > > > >
> > > > > > > References:
> > > > > > >
> > > > > > > [1] Dasgupta, Sanjoy. "Analysis of a greedy active learning strategy." Advances in neural information processing systems 17 (2005): 337-344.
> > > > > > >
> > > > > > > [2] Nowak, Robert. "Generalized binary search." 2008 46th Annual Allerton Conference on Communication, Control, and Computing. IEEE, 2008.
> > > > > > >
> > > > > > > [3] Nowak, Robert D. "The geometry of generalized binary search." IEEE Transactions on Information Theory 57.12 (2011): 7893-7906.
> > > > > > >
> > > > > > > [4]Mussmann, Stephen, and Percy Liang. "Generalized binary search for split-neighborly problems." International Conference on Artificial Intelligence and Statistics. PMLR, 2018.

---

> > > > > > > > ### Comment · Reviewer_NhnX · 2021-08-22
> > > > > > > > **further discussions**
> > > > > > > >
> > > > > > > > Thanks for the further comments! I personally think that the T+AL problem is quite interesting, and the results in the paper are non-trivial with some real insights. My further suggestions:
> > > > > > > >
> > > > > > > > - I really appreciate the discussions proposed by the authors. But many of them are not clearly and detailedly included in the paper. For example, 1) the teacher's effectiveness could be significantly influenced by the learner's state 2) the teaching complexity could be worsened by an adversarial learner. I think including such discussions could be even more important than providing only technical analysis.
> > > > > > > >
> > > > > > > > - I think it is important to explicitly analyze the $OPT^{T+AL}$ term since only doing this can tell us when T+AL can be effective. Even though this solution could be computationally hard to achieve, we can also provide approximation algorithms to them that have explicit sample complexity bounds without the dependence on $OPT^{T+AL}$. These bounds can be comparable to those AL sample complexity bounds.
> > > > > > > >
> > > > > > > > - I think the real examples of bad active learners are important. If they are provided, the significance of the paper can be strengthened significantly.
> > > > > > > >
> > > > > > > >
> > > > > > > > - For Q1: I think your explanations are reasonable. I understand that the real teaching sequences are influenced by the learner significantly no matter what $\gamma^g$ is. What I meant is that when $\gamma^g=1$, we could not observe the true effectiveness of teaching, since it is hidden in the $OPT^{T+AL}$ term. We can only observe the approximation power of the greedy teacher. For $\gamma^g=1$, we know that this power is dominated by the teacher since the learner is not quite "adversarial".
> > > > > > > >
> > > > > > > > Overall, my current evaluation is quite on the borderline. My score is based on the above unsatisfactory points. I do like many aspects of the paper, and if the above suggestions are well-treated, I would advocate that this is good work.

---

> > > > > > > > > ### Author Response · Authors · 2021-08-24
> > > > > > > > > **Analysis on OPT^{T+AL} and bad active learners**
> > > > > > > > >
> > > > > > > > > We are deeply appreciative of the reviewer’s continuous efforts to help us improve our paper! We take all comments seriously and try our best to address every raised concern. We sincerely hope that our following response can address your remaining concerns.
> > > > > > > > >
> > > > > > > > > In summary, in our revision we will include additional discussions on
> > > > > > > > >
> > > > > > > > > 1. the difference between $\gamma$ and $\gamma^g$;
> > > > > > > > > 2. how is the effectiveness of the teacher affected by the learner (i.e., the $\alpha$ parameter);
> > > > > > > > > 3. how can an adversarial learner affect the teaching complexity (i.e., the $\gamma^g$ parameter);
> > > > > > > > > 4. the following analysis on $\text{OPT}^{T+AL}$; and
> > > > > > > > > 5. bad active learners.
> > > > > > > > >
> > > > > > > > > Below please find our responses to the remaining concerns:
> > > > > > > > >
> > > > > > > > > **1. Analysis on $\text{OPT}^{T+AL}$**
> > > > > > > > > - Great suggestion! In our appendix, **we have shown that $\text{OPT}^{\text{T}}\leq \text{OPT}^{\text{T+AL}}\leq 2\cdot \text{OPT}^{\text{T}}$ in Lemma 1**, where $\text{OPT}^{\text{T}}$ is the teaching dimension. In general, the teaching dimension $\text{OPT}^{\text{T}}$ is much smaller than the **optimal** sample complexity of AL, i.e., $\text{OPT}^{\text{AL}}$. The following canonical example on learning $1$-d threshold function can illustrate this point:
> > > > > > > > > > Consider learning a 1D threshold classifier where the input distribution $P_X$ is uniform over the interval [0, 1], the true threshold is $\theta^*$, and the binary label is noiseless: $y := \theta^*(x) = \mathbb{1}_{x\geq \theta^*}$.
> > > > > > > > > > - For active learning, the best strategy is a binary search on the interval [0, 1]. With each query, the learner can remove half of the remaining interval since it can deduce that the threshold cannot be in that half. To achieve an $\epsilon$ generalization error, it requires $n \geq \mathcal{O}(\log(\epsilon^{-1}))$ samples.
> > > > > > > > > > - However, for machine teaching, the teacher only needs to pick two training items, one negative and the other positive, such that they are at most $\epsilon$ apart and contain $\theta^*$ in the middle. This results in constant sample complexity.
> > > > > > > > >
> > > > > > > > >   In the next, we give a more rigorous analysis on $\text{OPT}^{\text{T+AL}}$:
> > > > > > > > >
> > > > > > > > > - First of all, it is not too hard to show $\text{OPT}^{\text{T+AL}} \leq \text{OPT}^{\text{AL}}$. This result can be proved by the fact that the optimal teacher cannot perform worse than such a teacher, which mimics the query selection strategy of the active learner. This inequality implies that the optimal teacher is always helpful for the active learner!
> > > > > > > > >
> > > > > > > > > - We further argue that the gap between $\text{OPT}^{\text{T+AL}}$ and $\text{OPT}^{\text{AL}}$ can be *arbitrarily large*.  We refer to the work by Hanneke (2007) [1]. In theorem 1 (page 5), Henneke (2007) showed that $\text{OPT}^{\text{T}} \leq \text{OPT}^{\text{AL}} \leq \log|\mathcal{H}|\cdot \text{OPT}^{\text{T}}$. This indicates that the teaching dimension is a lower bound of the optimal sample complexity of AL. However, on the other hand, $\text{OPT}^{\text{AL}}$ can be as large as $\log|\mathcal{H}|\cdot \text{OPT}^{\text{T}}$. In contrast, $\text{OPT}^{\text{T+AL}}$ is upper bounded by $2\cdot\text{OPT}^{\text{T}}$. Therefore, for large hypothesis class (i.e., large $|\mathcal{H}|$), $\text{OPT}^{\text{AL}}$ can be significantly larger than $\text{OPT}^{\text{T+AL}}$, showing the great advantage of incorporating a teacher in active learning.
> > > > > > > > >
> > > > > > > > > - We thank the reviewer again for the great suggestion! We will make a table to highlight the relationship between $\text{OPT}^{\text{T+AL}}$, $\text{OPT}^{\text{AL}}$ and $\text{OPT}^{\text{T}}$. We believe such a table will better highlight the advantage of incorporating a teacher in active learning.
> > > > > > > > >
> > > > > > > > > **2. Real examples of bad active learners**
> > > > > > > > > - Thanks for the suggestion! We would like to point out that **we indeed have provided a stylized example in the appendix** (Section E, Proof of Remark 2), showing a concrete (pessimistic) scenario where a *greedy** teacher hurts the performance of an active learner.
> > > > > > > > >
> > > > > > > > > - Note that in our example, the active learner (i.e., a greedy GBS learner) is not an adversary by its nature; rather, the overhead of learning with a teacher is due to the fact that the myopic strategy of a greedy teacher distracts the active learner from sticking with the (originally optimal) binary search policy. We believe this counter-example clearly shows that the greedy teacher is not satisfactory. This example can be a useful sanity check for future research on proposing better teaching algorithms for active learners. We will highlight this example in the main paper.
> > > > > > > > >
> > > > > > > > > - Another possible case is that the active learner always selects similar examples as the teacher, i.e., redundant queries. But we do believe such active learners are rare in practice.
> > > > > > > > >
> > > > > > > > >
> > > > > > > > > We appreciate your continuous efforts in helping us improve this work! We sincerely hope that the above responses address your remaining concerns well, and are helpful for improving the rating. If you have any further questions/concerns/suggestions, please let us know!
> > > > > > > > >
> > > > > > > > >
> > > > > > > > > -----
> > > > > > > > > Reference
> > > > > > > > >
> > > > > > > > > [1] Hanneke, Steve. "Teaching dimension and the complexity of active learning." International Conference on Computational Learning Theory. Springer, Berlin, Heidelberg, 2007.

---

> > > > > > > > > > ### Comment · Reviewer_NhnX · 2021-08-25
> > > > > > > > > > **further discussions**
> > > > > > > > > >
> > > > > > > > > > Thanks for the feedback! I think adding these discussions would indeed make future readers obtain deeper understandings of the results in the paper.
> > > > > > > > > >
> > > > > > > > > > I think the paper would be a good one if all the above issues are addressed. After all, I would raise my score to six since I really think the T+AL problem is an interesting one. While I also feel that that there would be a significant change for the current paper to include all these discussions.
> > > > > > > > > >
> > > > > > > > > > Finally, I would like to express my thanks to the authors for the interesting discussions, from which I have also learned a lot.

---

> > > > > > > > > > > ### Author Response · Authors · 2021-08-25
> > > > > > > > > > > **Thanks a lot!**
> > > > > > > > > > >
> > > > > > > > > > > We genuinely thank the reviewer's tremendous efforts to help us improve this work! We really enjoyed the fruitful discussion with you!

---

### Official Review · Reviewer_PbTC · 2021-07-26

**Rating:** 5
**Confidence:** 3

**Summary:**

The paper proposes a teaching algorithm that uses contrastive examples to teach an active learner. Some performance guarantees are given for some special cases. Some empirical evidences are presented to support the results.

**Limitations And Societal Impact:**

They are discussed.

**Main Review:**

In general, I think the idea of using contrastive examples in machine teaching sounds interesting. However, what concerns me most is the significance of the problem. It seems a bit unmotivated to teach an active learner, since the active learner can already query samples from an oracle. If the teacher contains optimal information of the learner, then the problem seems to be meaningless. If the teacher is a black-box one, then the problem reduces to designing a new active learning algorithm. Either way, I find the motivation of the paper quite weak.

For the theoretical results, it doesn't seem to suggest that the teacher can indeed provably improve the sample complexity, since the guarantees are not sufficiently tight. The worst-case bound is also not informative for the teaching algorithm. The tightness of the bound (even under some assumptions) is important and also needs to be discussed. In general, I believe the theory part is not signficant enough.

The experiments show an incremental improvement over the standard active learner with some simulations on toy examples. The experiments look reasonable but not impressive. I will be interested to see some experiments on some realistic distributions (say MNIST, CIFAR-10/100).

============= post-rebuttal =============

Thanks for the response. My concerns are partially addressed. As a non-expert in machine teaching, I start to understand why making the passive learner active is useful and it also simulates the real-word education nicely. But I do think there is much room for improvement, such as stronger theoretical results and more practical experiments.

I will increase my score from 4 to 5 (I would pick 5.5 if that is an option). I think this paper is much more boarderline than I originally thought.



**Time Spent Reviewing:**

5

---

> ### Author Response · Authors · 2021-08-05
> **Response to Reviewer PbTC [1/2]**
>
> Thank you for your comments and suggestions! Regarding your questions and concerns, please refer to our responses below.
>
> ---
>
> **1. Significance of the problem**
>
> - The problem studied in our paper connects machine teaching and active learning. We not only allow the learner to ask queries but also incorporate the role of teacher to proactively provide explanatory information to guide the learner (i.e., contrastive examples). Such a cooperative learning and teaching protocol fully exploits the ability of the learner and the teacher. This contrasts with active learning, where the teacher is passive, and classic machine teaching, where the learner is passive. Therefore, we believe our studied problem is novel and important (this was also recognized by the other two Reviewers NhnX and j9HW).
>
> **2. Unmotivated to teach an active learner, since the active learner can already query samples from an oracle**
>
> - In our setting, the teacher is not simply an oracle, as it also **proactively** provides samples to the learner. In active learning, the learner needs to select the query samples by itself, and the oracle only provides the corresponding labels. This key difference makes our problem more challenging but also more useful. For a visual illustration of the difference between “classic machine teaching”, “active learning” and “teaching an active learner” are in Figure 1.
>
> - In addition, we would also like to highlight the difference between machine teaching and active learning with the following concrete example:
> > Consider learning a 1D threshold classifier where the input distribution $P_X$ is uniform over the interval [0, 1], the true threshold is $\theta^*$, and the binary label is noiseless: $y := \theta^*(x) = \mathbb{1}_{x\geq \theta^*}$.
> > - For active learning, the best strategy is binary search on the interval [0, 1]. With each query, the learner can remove half of the remaining interval since it can deduce that the threshold cannot be in that half. To achieve an $\epsilon$ generalization error, it requires $n \geq \mathcal{O}(\log(\epsilon^{-1}))$ samples.
> > - However, for machine teaching, the teacher only needs to pick two training items, one negative and the other positive, such that they are at most $\epsilon$ apart and contain $\theta^*$ in the middle. This results in a constant sample complexity. For a more detailed introduction on machine teaching, please refer to [1].
>
>
> **3. Black-box teacher**
>
> - Thanks for proposing the black-box setting. We assume that here “black-box teacher” refers to the scenario where the teacher does not have access to the learner’s full information, i.e, the teacher may not know the representation (function) and the hypothesis class of the learner. This is a natural future direction following up our paper. The high-level idea will be that the teacher should also actively learn the information of the learner’s representation, hypothesis class and query function. A related work is by Dasgupta et al., (2019) [2], where they extended the classic machine teaching to the black-box setting by formulating the problem as an online set cover problem.
>
>
> **4. The bound is not sufficiently tight; Teacher does not improve the sample complexity;**
>
> - We respectfully disagree that the bound is not sufficiently tight. Theorem 2 shows that the bound in Theorem 1 is non-vacuous. Meanwhile, to be noted, when the objective function $f(\cdot | q)$ is postfix-monotone and sequence submodular, then the bound in Theorem 1 can be simplified to $\alpha\cdot\mathcal{O}(\log(|\mathcal{H}|))\cdot \text{OPT}^{\text{T+AL}}$. This is the best bound for an $\alpha$-approximation greedy algorithm, which cannot be further improved under reasonable complexity-theoretic assumptions.
>
> - However, we would also like to note that, if the teacher is optimal, then the teacher is provable to improve the sample complexity. The proof is quite trivial by the fact that the optimal teacher cannot perform worse than such a teacher, which mimics the query selection strategy of the active learner.
>
> - Though our remark 2 shows that the greedy teacher is not always guaranteed to be helpful,
> empirically, we’ve demonstrated that greedy teacher *can* improve the sample complexity in most of the settings. Our results also indicate that the greedy teaching algorithm is not satisfactory when teaching an active learner. This provides crystal evidence for motivating future works to go beyond greedy algorithms, and our counter-example can be a useful sanity check for new teaching algorithms.

---

> > ### Author Response · Authors · 2021-08-05
> > **Response to Reviewer PbTC [2/2]**
> >
> > **5. Worst-case bound is not informative;**
> >
> > - The worst-case bound requires minimal assumptions, e.g., the distribution of the hypotheses and the samples, and the target hypothesis. We believe that the worst-case bound is meaningful, as every hypothesis in the version space can potentially be the target hypothesis. Also, in machine teaching, a vast amount of the literature considers the worst-case bound, namely the teaching dimension [1, 2, 3].  Lastly, our theoretical bound also aligns well with the empirical results.
> >
> >
> > **6. MNIST, CIFAR-10/100 experiments**
> >
> > - Thank you for the suggestion. Though our paper is theory-oriented, our intention for including an experiment section is to provide a proof-of-concept demonstration that the greedy teacher is helpful in practice and also better illustrate our theory. Running experiments on MNIST, CIFAR-10/100 requires introducing additional algorithmic twists to the teacher as well as the active learner (including sampling from the hypothesis distribution, dealing with noisy feedback, etc), and hence goes beyond the scope of this paper.
> >
> > ---
> > We hope that our responses can address your concerns well, and are helpful for improving the ratings. If you are not convinced of the significance of our work (as a theory-oriented package), or have any other concerns, please let us know! We will be happy to make further responses. We are looking forward to hearing back from you! Thank you again for the reviews!
> >
> > ---
> > References:
> >
> > [1] Zhu, X., Singla, A., Zilles, S., & Rafferty, A. N. (2018). An overview of machine teaching. arXiv preprint arXiv:1801.05927.
> >
> > [2] Dasgupta, S., Hsu, D., Poulis, S., & Zhu, X. (2019, May). Teaching a black-box learner. In International Conference on Machine Learning (pp. 1547-1555). PMLR.
> >
> > [3] Goldman, S. and Kearns, M. On the complexity of teaching. Journal of Computer and System Sciences, 50(1):20–31, 1995.

---

> ### Author Response · Authors · 2021-08-24
> **Response to Reviewer PbTC**
>
> Thanks for the updates! We sincerely thank the reviewer for recognizing the value of our work! We are glad to see that you increased the score, which is really encouraging to us! We take all comments seriously and try our best to address every raised concern.
>
>
> **1. Stronger theoretical results**
> - Thanks for the suggestion. We have conducted a rigorous analysis on the relationship between $\text{OPT}^{\text{T+AL}}, \text{OPT}^{\text{AL}}$ and $\text{OPT}^{\text{T}}$ in the response to Reviewer NhnX. We quoted the corresponding details below. We hope that you can find it helpful.
> > **1. Analysis on $\text{OPT}^{T+AL}$**
> > - Great suggestion! In our appendix, **we have shown that $\text{OPT}^{\text{T}}\leq \text{OPT}^{\text{T+AL}}\leq 2\cdot \text{OPT}^{\text{T}}$ in Lemma 1**, where $\text{OPT}^{\text{T}}$ is the teaching dimension. In general, the teaching dimension $\text{OPT}^{\text{T}}$ is much smaller than the **optimal** sample complexity of AL, i.e., $\text{OPT}^{\text{AL}}$.
> > In the next, we give a more rigorous analysis on $\text{OPT}^{\text{T+AL}}$.
> >  -First of all, it is not too hard to show $\text{OPT}^{\text{T+AL}} \leq \text{OPT}^{\text{AL}}$. This result can be proved by the fact that the optimal teacher cannot perform worse than such a teacher, which mimics the query selection strategy of the active learner. This inequality implies that the optimal teacher is always helpful for the active learner!
> > -We further argue that the gap between $\text{OPT}^{\text{T+AL}}$ and $\text{OPT}^{\text{AL}}$ can be *arbitrarily large*.  We refer to the work by Hanneke (2007) [1]. In theorem 1 (page 5), Henneke (2007) showed that $\text{OPT}^{\text{T}} \leq \text{OPT}^{\text{AL}} \leq \log|\mathcal{H}|\cdot \text{OPT}^{\text{T}}$. This indicates that the teaching dimension is a lower bound of the optimal sample complexity of AL. However, on the other hand, $\text{OPT}^{\text{AL}}$ can be as large as $\log|\mathcal{H}|\cdot \text{OPT}^{\text{T}}$. In contrast, $\text{OPT}^{\text{T+AL}}$ is upper bounded by $2\cdot\text{OPT}^{\text{T}}$. Therefore, for large hypothesis class (i.e., large $|\mathcal{H}|$), $\text{OPT}^{\text{AL}}$ can be significantly larger than $\text{OPT}^{\text{T+AL}}$, showing the great advantage of incorporating a teacher in active learning.
>
>
> - If you have other suggestions on improving the theory part, any follow-up discussions are welcome!
>
> Thanks again for your updates and suggestions!
>
> -----
> Reference
>
> [1] Hanneke, Steve. "Teaching dimension and the complexity of active learning." International Conference on Computational Learning Theory. Springer, Berlin, Heidelberg, 2007.

---

### Author Response · Authors · 2021-08-05
**To all reviewers**


We thank all the reviewers for their careful reviews and positive comments, including: (Reviewer PbTC) “the idea of using contrastive examples is interesting”, (Reviewer NhnX) “the paper has good originality and clarity” and (Reviewer j9HW) “the problem setting addressed is interesting and novel… the submission seems technically sound, and the claims seem to be well supported”.

Our work is theory-oriented rather than application-oriented. In our paper, we first proposed a novel interaction framework that integrates the idea of machine teaching and active learning. This framework is more flexible and practical than the classical machine teaching framework, where the learner is passive. Also, the theoretical analysis is more challenging in our setting, as the *order* of teaching examples can affect the teaching performance. This is not the case for classical machine teaching as considered in the seminal work of Goldman & Kearns (1995) [1] (cf. Line 47-51 of our draft), as well as in the body of machine teaching literature concerning the teaching complexity for version space learners, where the problem can be reduced to a *set*-cover problem.

---

Based on the reviews, we would like to highlight a few additional contributions:

On the theoretical side:

- We studied a novel machine teaching setting with a natural teacher-learner interaction protocol. We first provided a general bound for the sample complexity of greedy teaching with arbitrary active learners (Theorem 1). We further showed that the bound in Theorem 1 is non-vacuous (Theorem 2). To be noted, when the objective function $f(\cdot | q)$ is postfix-monotone and sequence submodular, the bound in Theorem 1 can be simplified to $\alpha\cdot\mathcal{O}(\log(|\mathcal{H}|))\cdot \text{OPT}^{\text{T+AL}}$. This is the best bound for an $\alpha$-approximation greedy algorithm, which cannot be further improved in general under reasonable complexity-theoretic assumptions.

- Built upon Theorem 1, we further derived the bound for the sample complexity of GBS learners with a greedy teacher (Theorem 3). We believe it is a novel contribution which connects to the classical structural assumptions studied in the active learning literature, such as the k-neighborly condition introduced in Nowak (2008). In addition, we also constructed a realizable example, where the sample complexity of GBS with a greedy teacher is worse than that of GBS alone (Remark 2). This highlights the limitation of greedy algorithms in some specific cases. We believe this result encourages future works to go beyond greedy algorithms, and our counterexample is a good start point for designing better teaching algorithms for active learners.

On the empirical side:

- Though our work has a theoretical focus, we also provided carefully designed numerical experiments on two datasets. Our empirical results demonstrated the interesting tradeoff between different constraints, and showed the advantage of incorporating the role of greedy teacher in active learning. Furthermore, we also empirically validated the proposed theorems and showed that they are consistent with the empirical results.

-----

We believe machine teaching + active learning is an exciting but relatively under-explored direction. This setting is more practical than classic machine teaching (machine teaching + passive learning), and also more sample efficient than active learning. Our work made the first step in this direction with both theory and experiments.

We are looking forward to hearing back from you!

----
Reference:

[1] Goldman, S. and Kearns, M. On the complexity of teaching. Journal of Computer and System Sciences, 50(1):20–31, 1995.

---

### Decision · Program_Chairs · 2021-09-27

**Decision:**

Accept (Poster)

**Comment:**

This paper proposes a novel learning setup, provides some initial sample and query complexity analysis as well as an empirical evaluation on a synthetic dataset. The authors propose a merge of active learning and machine teaching. In their framework, a learner gets to query an example, and the teacher responds with a label and an additional piece of information. Here, the teacher also provides a "contrastive example", that is an example that is similar to the query but is from a different label class or one that is dissimilar but from the same label class.

The reviewers appreciated the clarity and novelty in this submission. Some concerns regarding correctness were raised, but resolved during the discussion phase. The reviewers also raised some concerns about motivating this particular setup.

Overall, this seems to be a solid contribution, that provides a novel framework and initial analysis. The work will likely be of interest to researchers in the neurips community who focus on active learning, query learning, or machine teaching.